# Data driven semi-supervised learning

**Maria-Florina Balcan**
School of Computer Science
Carnegie Mellon University
Pittsburgh, PA 15213
`ninamf@cs.cmu.edu`

**Dravyansh Sharma**
Department of Computer Science
Carnegie Mellon University
Pittsburgh, PA 15213
`dravyans@cs.cmu.edu`

## Abstract

We consider a novel data driven approach for designing semi-supervised learning algorithms that can effectively learn with only a small number of labeled examples. We focus on graph-based techniques, where the unlabeled examples are connected in a graph under the implicit assumption that similar nodes likely have similar labels. Over the past two decades, several elegant graph-based semi-supervised learning algorithms for inferring the labels of the unlabeled examples given the graph and a few labeled examples have been proposed. However, the problem of how to create the graph (which impacts the practical usefulness of these methods significantly) has been relegated to heuristics and domain-specific art, and no general principles have been proposed. In this work we present a novel data driven approach for learning the graph and provide strong formal guarantees in both the distributional and online learning formalizations. We show how to leverage problem instances coming from an underlying problem domain to learn the graph hyperparameters for commonly used parametric families of graphs that provably perform well on new instances from the same domain. We obtain low regret and efficient algorithms in the online setting, and generalization guarantees in the distributional setting. We also show how to combine several very different similarity metrics and learn multiple hyperparameters, our results hold for large classes of problems. We expect some of the tools and techniques we develop along the way to be of independent interest, for data driven algorithms more generally.

## 1 Introduction

In recent years machine learning has found gainful application in diverse domains. A major bottleneck of the currently used approaches is the heavy dependence on expensive labeled data. Advances in cheap computing and storage have made it relatively easier to store and process large amounts of unlabeled data. Therefore, an important focus of the present research community is to develop general domain-independent methods to learn effectively from the unlabeled data, along with a small amount of labels. Achieving this goal would significantly elevate the state-of-the-art machine intelligence, which currently lags behind the human capability of learning from a few labeled examples. Our work is a step in this direction, and provides algorithms and guarantees that enable fundamental techniques for semi-supervised learning to provably adapt to problem domains.

Graph-based approaches have been popular for learning from unlabeled data for the past two decades [Zhu and Goldberg, 2009]. Labeled and unlabeled examples form the graph nodes and (possibly weighted) edges denote the feature similarity between examples. The graph therefore captures how each example is related to other examples, and by optimizing a suitably regularized objective over it one obtains an efficient discriminative, nonparametric method for learning the labels. There are several well-studied ways to define and regularize an objective on the graph [Chapelle et al., 2010],

35th Conference on Neural Information Processing Systems (NeurIPS 2021).

Table 1: Optimization objectives for graph-based SSL. $D_{ij} := \mathbb{I}[i = j] \sum_k W_{ik}, \mathcal{L} := D^{-1/2}(D - W)D^{-1/2}$ and the objective is $l(f) = \alpha \sum_{u \in L}(f(u) - y_u)^2 + \beta H(f, W) + \gamma \|f\|^2$.

| Algorithm | $(\alpha, \beta, \gamma)$ | $H(f, W), \|\cdot\|$ | Constraints on $f$ |
|---|---|---|---|
| Mincut | $(\infty, 1, 0)$ | $f^T(D - W)f$ | $f \in \{0, 1\}^n$ |
| Harmonic function | $(\infty, 1, 0)$ | $f^T(D - W)f$ | $f \in [0, 1]^n$ |
| Normalized cut | $(\infty, 1, 0)$ | $f^T(D - W)f$ | $f^T\mathbf{1} = 0, f^Tf = n^2, f \in [0, 1]^n$ |
| Label propagation | $(1, \mu, 1)$ | $f^T\mathcal{L}f, \|\cdot\|_2$ | $f \in [0, 1]^n$ |

and all yield comparable results which strongly depend on the graph used. A general formulation is described as follows, variations on which are noted under related work.

**Problem formulation** Given sets $L$ and $U$ of labeled and unlabeled examples respectively, and a similarity metric $d$ over the data, the goal is to use $d$ to extrapolate labels in $L$ to $U$. A graph $G$ is constructed with $L + U$ as the nodes and weighted edges $W$ with $w(u, v) = g(d(u, v))$ for some $g : \mathbb{R}_{\geq 0} \to \mathbb{R}_{\geq 0}$. We seek labels $f(\cdot)$ for nodes $u$ of $G$ which minimize a regularized loss function $l(f) = \alpha \sum_{v \in L} \hat{l}(f(v), y_v) + \beta H(f, W) + \gamma \|f\|^2$, under some constraints on $f$. The objective $H$ captures the *smoothness* (regularization) induced by the graph (see Table 1 for examples) and $\hat{l}(f(v), y_v)$ is the misclassification loss (computed here on labeled examples).

The graph $G$ takes a central position in this formulation. However, the majority of the research effort on this problem has focused on how to design and optimize the regularized loss function $l(f)$, the effectiveness of which crucially depends on $G$. There is no known principled study on how to build $G$ and prior work largely treats this as a domain-specific art [Chapelle et al., 2010]. Is it possible to acquire the required domain expertise, without involving human experts? In this work we provide an affirmative answer by formulating graph selection as *data-driven design*. More precisely, we are required to solve not only one instance, but multiple instances of the underlying algorithmic problem that come from the same domain [Gupta and Roughgarden, 2016, Balcan, 2020]. We show learning a near-optimal graph over commonly used infinite parameterized families is possible in both online and distributional settings. In the process we generalize and extend data-driven learning techniques, and obtain practical methods to build the graphs with strong guarantees. In particular, we show how the techniques can learn several parameters at once, and also learn a broader class of parameters than previously known.

**Our contributions and key challenges.** We present a first theoretically grounded work for graph-based learning from limited labeled data, while extending general data-driven design techniques.

*Data-driven algorithm design.* Firstly, for one dimensional loss functions, we show a novel structural result which applies when discontinuities (for loss as function of the algorithm parameter) occur along roots of exponential polynomials with random coefficients with bounded joint distributions (previously known only for algebraic polynomials in Balcan et al. [2020b]). This is crucial for showing learnability in the Gaussian graph kernels setting. Secondly, Balcan et al. [2020b] only applies when the discontinuities occur along algebraic curves with random coefficients in just two dimensions. By a novel algebraic and learning theoretic argument we are able to analyze higher (arbitrary constant number of) dimensions, making the technique much more generally applicable.

*Semi-supervised learning.* We examine commonly used parameterized graph families, denoted by general notation $G(\rho)$, where $\rho$ corresponds to a semi-supervised learning algorithm. We consider online and distributional settings, providing efficient algorithms to obtain low regret and low error respectively for learning $\rho$. Most previously studied settings involve polynomially many discontinuities for loss as function of the hyperparameter $\rho$ on a fixed instance, implying efficient algorithms, which may not be the case for our setting. To resolve this, we describe efficient semi-bandit implementations, and in particular introduce a novel min-cut and flow recomputation algorithm on graphs with continuously changing edge weights which may be of independent interest. For the distributional setting, we provide asymptotically tight bounds on the pseudodimension of the parameter learning problem. Our lower bounds expose worst case challenges, and involve precise constructions of problem instances by setting node similarities which make assigning labels provably hard.

Our techniques are extremely general and are shown to apply for nearly all combinations of optimization algorithms (Table 1) and parametric graph families (Definition 1).

**Related work**  *Semi-supervised learning* is a paradigm for learning from labeled and unlabeled data (Zhu and Goldberg [2009]). It resembles human learning behavior more closely than fully supervised and fully unsupervised models (Zhu et al. [2007], Gibson et al. [2013]). A popular approach for semi-supervised learning is to optimize a graph-based objective. Several methods have been proposed to predict labels given a graph including $st$-mincuts (Blum and Chawla [2001]), soft mincuts that optimize a harmonic objective (Zhu et al. [2003]), label propagation (Xiaojin and Zoubin [2002]), and many more (Shi and Malik [2000], Belkin et al. [2006]). All algorithms have comparable performance provided the graph $G$ encodes the problem well [Zhu and Goldberg, 2009]. However, it is not clear how to create the graph itself on which the extensive literature stands, barring some heuristics (Zhu et al. [2005], Zemel and Carreira-Perpiñán [2004]). Sindhwani et al. [2005] construct *warped* kernels aligned with the data geometry, but the performance may vary strongly with warping and it is not clear how to optimize over it. We provide the first techniques that yield provably near-optimal graphs.

Gupta and Roughgarden [2016, 2017] define a formal learning framework for selecting algorithms from a family of heuristics or setting hyperparameters. It is further developed by Balcan et al. [2017] and noted as a fundamental algorithm design perspective [Blum, 2020]. It has been successfully applied to several combinatorial problems like integer programming and clustering [Balcan et al., 2018a, 2019, 2018c] and for giving powerful guarantees like adversarial robustness, adaptive learning and differential privacy [Balcan et al., 2018b, 2020a,c, Vitercik et al., 2019, Balcan et al., 2020e,d]. Balcan et al. [2018b, 2020b] introduce general data-driven design techniques under some smoothness assumptions. We extend the techniques to significantly broader problem settings, and investigate the structure of graph-based label learning formulation to apply the new techniques.

## 2  Setup and definitions

We are given some unlabeled points $U \subset \mathcal{X}$ and labeled points $L \subset \mathcal{X} \times \mathcal{Y}$, such that $|L| + |U| = n$. One constructs a graph $G$ by placing (possibly weighted) edges $w(u, v)$ between pairs of data points $u, v$ which are 'similar', and labels for the unlabeled examples are obtained by optimizing some graph-based score. We have an oracle $O$ which on querying provides us the labeled and unlabeled examples, and we need to pick graph $G(\rho)$ from some family $\mathcal{G}$ of graphs, parameterized using a parameter $\rho \in \mathcal{P}$. We commit to using some graph labeling algorithm $A(G, L, U)$ (abbreviated as $A_{G,L,U}$) which provides labels for examples in $U$, and we should pick a $\rho$ such that $A(G(\rho), L, U)$ results in small error in its predictions on $U$. More formally, for a loss function $l : \mathcal{Y} \times \mathcal{Y} \to [0, 1]$ and a target labeling $\tau : U \to \mathcal{Y}$, we need to find $\mathrm{argmin}_{\rho \in \mathcal{P}} \, l_{A(G(\rho),L,U)} := \sum_U l(A_{G(\rho),L,U}(u), \tau(u))$.

We will now describe some graph families $\mathcal{G}$ and algorithms $A_{G,L,U}$. We assume there is a feature based *similarity function* $d : \mathcal{X} \times \mathcal{X} \to \mathbb{R}_{\geq 0}$, a metric which monotonically captures pairwise similarity. Commonly used parametric methods to build a graph using the similarity function follow.

**Definition 1.** *Graph kernels.*[1]

*a) Threshold graph, $G(r)$. Parameterized by a threshold $r$, we set $w(u, v) = \mathbb{I}[d(u, v) \leq r]$.*

*b) Polynomial kernel, $G(\tilde{\alpha})$. $w(u, v) = (\tilde{d}(u, v) + \tilde{\alpha})^d$ for fixed degree $d$, parameterized by $\tilde{\alpha}$.*

*c) Gaussian RBF or exponential kernel, $G(\sigma)$. $w(u, v) = e^{-d(u,v)^2/\sigma^2}$, parameterized by $\sigma$.*

**Remark 1.** *Another popular family of graphs used in practice is the $k$ nearest neighbor graphs, where $k \in \{0, 1, \dots, n-1\}$, $n$ is the number of nodes in the graph, is the parameter. Even though $k$-NN graphs may result in different graphs the ones considered in the paper, learning how to build an optimal graph over the algorithm family $G(k)$ is much simpler. Online learning of the parameter $k$ in this setting can be recognized as an instance of learning with experts advice for a finite hypothesis class (Section 3.1 of Shalev-Shwartz et al. [2011]), where an upper bound of $O(\sqrt{T \log n})$ is known for the Weighted Majority algorithm. Online-to-batch conversion provides generalization guarantees in the distributional setting (Section 5 of Shalev-Shwartz et al. [2011]). We remark that our algorithm families need more sophisticated analysis due to continuous ranges of the algorithm parameters.*

---

[1]With some notational abuse, we have $d$ as the integer polynomial degree, and $d(\cdot, \cdot)$ as the similarity function. Common choices are setting $d(u, v)$ as the Euclidean norm and $\tilde{d}(u, v)$ as the dot product when $u, v \in \mathbb{R}^n$.

The threshold graph adds (unweighted) edges to $G$ only when the examples are closer than some $r \geq 0$. We refer to this setting by the *unweighted graph* setting, and the others by the *weighted graph* setting. The similarity function $\tilde{d}(u, v)$ in Definitions 1b increases monotonically with similarity of examples (as opposed to the other two). Once the graph is constructed using one of the above kernels, we can assign labels using some algorithm $A_{G,L,U}$. A popular, effective approach is to optimize a quadratic objective $\frac{1}{2} \sum_{u,v} w(u,v)(f(u) - f(v))^2$. $f$ may be discrete, $f(u) \in \{0, 1\}$ corresponds to finding a mincut separating the oppositely labeled vertices [Blum and Chawla, 2001], or $f \in [0, 1]$ may be continuous and we can round $f$ to obtain the labels [Zhu et al., 2003]. These correspond to the *mincut* and *harmonic function* algorithms respectively from Table 1.

We also need some well-known definitions from prior work (Appendix A). In particular, we use *dispersion* from [Balcan et al., 2020b]. The sequence of random loss functions $l_1, \ldots, l_T$ is $\beta$-*dispersed* for the Lipschitz constant $L$ if, for all $T$ and for all $\epsilon \geq T^{-\beta}$, $\mathbb{E}\left[\max_{\rho, \rho' \in \mathcal{C}, \|\rho - \rho'\|_2 \leq \epsilon} |\{t \in [T] \mid l_t(\rho) - l_t(\rho') > L \|\rho - \rho'\|_2\}|\right] \leq \tilde{O}(\epsilon T)$.

## 3   New general dispersion-based tools for data-driven design

We present new general tools for analyzing data-driven algorithms. Our new tools apply to a very broad class of algorithm design problems, for which we derive sufficient *smoothness* conditions to infer dispersion of a random sequence of problems, i.e. the algorithmic performance as a function of the algorithm parameters is dispersed. Recall that dispersion, roughly speaking, captures the rate at which discontinuities concentrate in any region of the domain. Balcan et al. [2020b] provide a general tool for verifying dispersion if non-Lipschitzness occurs along roots of (algebraic) polynomials in one and two dimensions. We improve upon their results in two major ways.

Our first result is that dispersion for one-dimensional loss functions follows when the points of discontinuity occur at the roots of exponential polynomials if the coefficients are random, lie within a finite range, and are drawn according to a bounded joint distribution. The key idea is use algebraic arguments and Taylor series approximation to show that for any small interval containing roots of the random exponential polynomial, the corresponding sets of coefficients lie on $n - 1$ dimensional linear subspaces with a probability measure proportional to the length of the interval (Appendix C.3).

**Theorem 2.** *Let $\phi(x) = \sum_{i=1}^{n} a_i e^{b_i x}$ be a random function, such that coefficients $a_i$ are real and of magnitude at most $R$, and distributed with joint density at most $\kappa$. Then for any interval $I$ of width at most $\epsilon$, $P(\phi$ has a zero in $I) \leq \tilde{O}(\epsilon)$ (dependence on $b_i, n, \kappa, R$ suppressed).*

*Proof Sketch.* For $n = 1$ there are no roots, so assume $n > 1$. Suppose $\rho$ is a root of $\phi(x)$. Then $\mathbf{a} = (a_1, \ldots, a_n)$ is orthogonal to $\varrho(\rho) = (e^{b_1 \rho}, \ldots, e^{b_n \rho})$ in $\mathbb{R}^n$. For a fixed $\rho$, the set $S_\rho$ of coefficients $\mathbf{a}$ for which $\rho$ is a root of $\phi(y)$ lie along an $n - 1$ dimensional linear subspace of $\mathbb{R}^n$. Now $\phi$ has a root in any interval $I$ of length $\epsilon$, exactly when the coefficients lie on $S_\rho$ for some $\rho \in I$. The desired probability is therefore upper bounded by $\max_\rho \text{VOL}(\cup S_y \mid y \in [\rho - \epsilon, \rho + \epsilon]) / \text{VOL}(S_y \mid y \in \mathbb{R})$ which we will show to be $\tilde{O}(\epsilon)$. The key idea is that if $|\rho - \rho'| < \epsilon$, then $\varrho(\rho)$ and $\varrho(\rho')$ are within a small angle $\theta_{\rho, \rho'} = \tilde{O}(\epsilon)$ for small $\epsilon$ (the probability bound is vacuous for large $\epsilon$). But any point in $S_\rho$ is at most $\tilde{O}(\theta_{\rho, \rho'})$ from a point in $S_{\rho'}$, which implies the desired bound. $\square$

We further go beyond single-parameter discontinuities, which occur as points along a line to general small dimensional parameter spaces $\mathbb{R}^p$, where discontinuties can occur along algebraic hypersurfaces. We employ tools from algebraic geometry to establish a bound on shattering of algebraic hypersurfaces by axis-aligned paths (Theorem 3), which implies dispersion using a VC dimension based argument (Theorem 4). Our result is a first general sufficient condition for dispersion for any constant number $p$ of parameters, and applies to a broad class of algorithm families. Full proofs are in Appendix C.4.

**Theorem 3.** *There is a constant $k$ depending only on $d$ and $p$ such that axis-aligned line segments in $\mathbb{R}^p$ cannot shatter any collection of $k$ algebraic hypersurfaces of degree at most $d$.*

*Proof Sketch.* Let $\mathcal{C}$ denote a collection of $k$ algebraic hypersurfaces of degree at most $d$ in $\mathbb{R}^p$. We say that a subset of $\mathcal{C}$ is *hit* by a line segment if the subset is exactly the set of curves in $\mathcal{C}$ which intersect the segment. We can upper bound the subsets of $\mathcal{C}$ hit by line segments in a fixed axial direction $x$ in two steps. Along a fixed line, Bezout's Theorem bounds the number of intersections

and therefore subsets hit by different line segments. Using the Tarski–Seidenberg Theorem, the lines along $x$ can be shown to belong to equivalence classes corresponding to cells in the cylindrical algebraic decomposition of the projection of the hypersurfaces, orthogonal to $x$. Finally, this extends to axis-aligned segments by noting they may hit only $p$ times as many subsets. $\qquad\square$

**Theorem 4.** *Let $l_1, \dots, l_T : \mathbb{R}^p \to \mathbb{R}$ be independent piecewise L-Lipschitz functions, each having discontinuities specified by a collection of at most $K$ algebraic hypersurfaces of bounded degree. Let $L$ denote the set of axis-aligned paths between pairs of points in $\mathbb{R}^p$, and for each $s \in L$ define $D(T, s) = |\{1 \le t \le T \mid l_t \text{ has a discontinuity along } s\}|$. Then we have $\mathbb{E}[\sup_{s \in L} D(T, s)] \le \sup_{s \in L} \mathbb{E}[D(T, s)] + O(\sqrt{T \log(TK)})$.*

## 4 Learning the graph online

We will warm up this section with a simple example demonstrating the need for and challenges posed by the problem of learning how to build a good graph from data. We consider the setting of learning thresholds for unweighted graphs (Definition 1a). We give a simple demonstration that in a single instance *any threshold* may be optimal for labelings consistent with graph smoothness assumptions, therefore providing motivation for the learning in our setting. Our construction (depicted in Figure 1) captures the intuition that any unlabeled point may get weakly connected to examples from one class for a small threshold but may get strongly connected to another class as the threshold is increased to a larger value. Therefore depending on the unknown true label either threshold may be optimal or suboptimal, and it makes sense to learn the correct value through repeated problem instances.

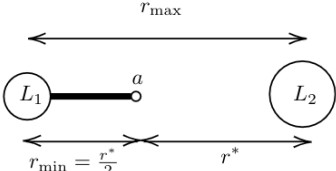

Figure 1: $G(r)$ connects $a$ to nodes in $L_1$ for $r_{\min} \le r < r^*$. $|L_1| < |L_2|$.

**Theorem 5.** *Let $r_{\min}$ denote the smallest value of threshold $r$ for which every unlabeled node of $G(r)$ is reachable from some labeled node, and $r_{\max}$ be the smallest value of threshold $r$ for which $G(r)$ is the complete graph. There exists a data instance $(L, U)$ such that for any $r_\zeta = \zeta r_{\min} + (1 - \zeta) r_{\max}$ for $\zeta \in (0, 1)$, there exists a set of labelings $\mathcal{U}$ of the unlabeled points such that for some $U_\zeta, \bar{U}_\zeta \in \mathcal{U}$, $r_\zeta$ minimizes $l_{A(G(r), L, U_\zeta)}$ but not $l_{A(G(r), L, \bar{U}_\zeta)}$.*

### 4.1 Dispersion and online learning

We consider the problem of learning the graph online. In this setting, we are presented with instances of the problem online and want to learn the best value of the parameter $\rho$ while making predictions. For now, we assume we get all the labels for past instances which may be used to determine the loss for any $\rho$ (*full information*). At time $t \in [T]$ we predict $\rho_t \in \mathcal{P}$ (the parameter space) based on labeled and unlabeled examples $(L_i, U_i), i \in [t]$ and past labels $\tau(u)$ for each $u \in U_j, j < t$ and seek to minimize regret $R_T := \sum_{t=1}^{T} l_{A(G(\rho_t), L_t, U_t)} - \min_{\rho \in \mathcal{P}} \sum_{t=1}^{T} l_{A(G(\rho), L_t, U_t)}$.

A key difficulty in the online optimization for our settings is that the losses are discontinuous functions of the graph parameters $\rho$. We can efficiently solve this problem if we can show that the loss functions are dispersed, in fact $\frac{1}{2}$-dispersed functions may be learned with $\tilde{O}(\sqrt{T})$ regret (Balcan et al. [2018b, 2020c]). Algorithm 1 adapts the general algorithm of Balcan et al. [2018b] to data-driven graph-based learning and achieves low regret for dispersed functions. Recall that dispersion roughly says that the discontinuities in the loss function are not too concentrated. We will exploit an assumption that the embeddings are approximate, so small random perturbations to the distance metric will likely not affect learning. This mild distributional assumption allows us to show that Algorithm 1 learns $\rho$.

**Algorithm 1** Data-driven Graph-based SSL

---

1: **Input:** Graphs $G_t$ with labeled and unlabeled nodes $(L_t, U_t)$, node similarities $d(u,v)_{u,v \in L_t \cup U_t}$.

2: **Hyperparameter:** step size parameter $\lambda \in (0,1]$.
3: **Output:** Graph parameter $\rho_t$ for times $t = 1, 2, \ldots, T$.
4: Set $w_1(\rho) = 1$ for all $\rho \in \mathbb{R}_{\geq 0}$.
5: **for** $t = 1, 2, \ldots, T$ **do**
6:     Sample $\rho$ with probability $p_t(\rho) = \frac{w_t(\rho)}{W_t}$, output as $\rho_t$, where $W_t := \int_C w_t(\rho) d\rho$.
7:     Compute average loss function $l_t(\rho) = \frac{1}{|U_t|} \sum_{u \in U} l(A_{G_t(\rho), L_t, U_t}(u), \tau(u))$.
8:     For each $\rho \in C$, set $w_{t+1}(\rho) = e^{\lambda u_t(\rho)} w_t(\rho)$, where $u_t(\rho) = 1 - l_t(\rho) \in [0,1]$.

---

### 4.1.1 Dispersion of the loss functions.

We first show dispersion for the unweighted graph family, with threshold parameter $r$. Here dispersion follows from a simple assumption that the distance $d(u,v)$ for any pair of nodes $u, v$ follows a $\kappa$-bounded distribution[2], and observing that discontinuities of the loss (as a function of $r$) must lie on the set of distances $d(u,v)$ in the samples (for any optimization algorithm). Using a VC dimension argument on the loss sequence we show (Appendix C.1).

**Theorem 6.** *Let $l_1, \ldots, l_T : \mathbb{R} \to \mathbb{R}$ denote an independent sequence of losses as a function of parameter $r$, when the graph is created using a threshold kernel $w(u,v) = \mathbb{I}[d(u,v) \leq r]$ and labeled by applying any algorithm on the graph. If $d(u,v)$ follows a $\kappa$-bounded distribution for any $u, v$, the sequence is $\frac{1}{2}$-dispersed, and the regret of Algorithm 1 is $\tilde{O}(\sqrt{T})$.*

We also show dispersion for weighted graph kernels, but under slightly stronger assumptions. We assume that distances $d(u,v)$ are jointly $\kappa$-bounded on a closed and bounded support. The plan is show that if the similarity function is smooth, then the discontinuities lie along roots of a polynomial with random finite coefficients with a $\kappa'$-bounded joint distribution, and use results for dispersion analysis from Balcan et al. [2020b]. We establish the following theorem (proof in Appendix C.2).

**Theorem 7.** *Let $l_1, \ldots, l_T : \mathbb{R} \to \mathbb{R}$ denote an independent sequence of losses as a function of $\tilde{\alpha}$, for graph with edges $w(u,v) = (\tilde{d}(u,v) + \tilde{\alpha})^d$ labeled by optimizing the quadratic objective $\sum_{u,v} w(u,v)(f(u) - f(v))^2$. If $\tilde{d}(u,v)$ follows a $\kappa$-bounded distribution with a closed and bounded support, the sequence is $\frac{1}{2}$-dispersed, and the regret of Algorithm 1 may be upper bounded by $\tilde{O}(\sqrt{T})$.*

*Proof Sketch.* The solution of the quadratic objective is given by $f_U = (D_{UU} - W_{UU})^{-1} W_{UL} f_L$. The key technical challenge is to show that for any $u \in U$, $f(u) = 1/2$ is a polynomial equation in $\tilde{\alpha}$ with degree at most $nd$, and coefficients that are jointly $K\kappa$-bounded, where $K$ is a constant that only depends on $d$ and the support of $\tilde{d}(u,v)$. Therefore the labeling, and consequently also the loss function, may only change when $\tilde{\alpha}$ is a root of one of $|U|$ polynomials of degree at most $dn$. The dispersion result is now a simple application of results from Balcan et al. [2020b]. $\square$

**Remark 2.** *Theorem 6 applies to all objectives in Table 1, and Theorem 7 extends to all except the mincut. We can also extend the analysis to obtain similar results when using the exponential kernel $w(u,v) = e^{-||u-v||^2/\sigma^2}$. The results of Balcan et al. [2020b] no longer directly apply as the points of discontinuity are no longer roots of polynomials, and we need to analyze points of discontinuities of exponential polynomials, i.e. $\phi(x) = \sum_{i=1}^k a_i e^{b_i x}$ (See Section 3 and Appendix C.3).*

**Remark 3** (Extension to local and global classification Zhou et al. [2004])**.** *Above results can be extended to the classification algorithm used in Zhou et al. [2004]. The key observation is that the labels are given by a closed-form matrix, $f^* = (I - \alpha D^{-1/2} W D^{1/2})Y$ or $f^* = (D - \alpha W)Y$ (for the two variants considered). For threshold graphs $G(r)$, the regret bound in Theorem 6 applies to any classification algorithm. Extension to polynomial kernels $G(\tilde{\alpha})$ is described below. For fixed $\alpha$ (in the notation of Zhou et al. [2004], in expression for $f^*$ above), the discontinuities in the loss as a function of the parameter $\tilde{\alpha}$ lie along roots of polynomials in the parameter $\tilde{\alpha}$ and therefore the same proof as Theorem 7 applies (essentially we get polynomial equations with slightly different but still*

---

[2]A density function $f : \mathbb{R} \to \mathbb{R}$ is $\kappa$-bounded if $\max_{x \in \mathbb{R}}\{f(x)\} \leq \kappa$. $\mathcal{N}(\mu, \sigma)$ is $\frac{1}{2\pi\sigma}$-bounded for any $\mu$.

*K-bounded coefficients). On the other hand, if we consider $\alpha$ as another graph parameter, we can still learn the kernel parameter $\tilde{\alpha}$ together with $\alpha$ by applying Theorem 18 and Theorem 4 (instead of Theorem 19) in the proof of Theorem 7.*

### 4.1.2 Combining several similarity measures.

Multiple natural metrics often existin multimodal semi-supervised learning [Balcan et al., 2005]. Different metrics may have their own advantages and issues and often a weighted combination of metrics, say $\sum_i \rho_i d_i(\cdot, \cdot)$, works better than any individual metric. The combination weights $\rho_i$ are additional graph hyperparameters. A combination of metrics is known to boost performance theoretically and empirically for linkage-based clustering [Balcan et al., 2019]. However the argument therein crucially relies on the algorithm depending on relative distances and not the actual values, and therefore does not extend directly to our setting. We develop a first general tool for analyzing dispersion for multi-dimensional parameters (Section 3), which implies the multi-parameter analogue of Theorem 7, stated below. See Appendix C.4 for proof details.

**Theorem 8.** *Let $l_1, \ldots, l_T : \mathbb{R}^p \to \mathbb{R}$ denote an independent sequence of losses as a function of parameters $\rho_i, i \in [p]$, when the graph is created using a polynomial kernel $w(u, v) = (\sum_{i=1}^{p-1} \rho_i \tilde{d}(u, v) + \rho_p)^d$ and labeled by optimizing the quadratic objective $\sum_{u,v} w(u, v)(f(u) - f(v))^2$. If $\tilde{d}(u, v)$ follows a $\kappa$-bounded distribution with a closed and bounded support, the sequence is $\frac{1}{2}$-dispersed, and the regret of Algorithm 1 may be upper bounded by $\tilde{O}(\sqrt{T})$.*

### 4.1.3 Semi-bandit setting and efficient algorithms.

Online learning with full information is usually inefficient in practice since it involves computing and working with the entire domain of hyperparameters. For our setting in particular this is computationally infeasible for weighted graphs since the number of pieces (in loss as a piecewise constant function of the parameter) may be exponential in the worst case (see Section 5). Fortunately we have a workaround provided by Balcan et al. [2020b] where dispersion implies learning in a semi-bandit setting as well. This setting differs from the full information online problem as follows. In each round as we select the parameter $\rho_i$, we only observe losses for a single interval containing $\rho_i$ (as opposed to the entire domain). We call the set of these observable intervals the *feedback set*, and these provide a partition of the domain.

---

**Algorithm 2** Efficient Data-driven Graph-based SSL

---

1: **Input:** Graphs $G_t$ with labeled and unlabeled nodes $(L_t, U_t)$, node similarities $d(u, v)_{u,v \in L_t \cup U_t}$.

2: **Hyperparameter:** step size parameter $\lambda \in (0, 1]$.
3: **Output:** Graph parameter $\rho_t$ for times $t = 1, 2, \ldots, T$.
4: Set $w_1(\rho) = 1$ for all $\rho \in C$
5: **for** $t = 1, 2, \ldots, T$ **do**
6:     Sample $\rho$ with probability $p_t(\rho) = \frac{w_t(\rho)}{W_t}$, output as $\rho_t$, where $W_t := \int_C w_t(\rho) d\rho..$
7:     Compute the feedback set $A^{(t)}(\rho)$ containing $\rho_t$.
    For example, for the min-cut objective use Algorithm 3 (Appendix C.5.1) and set $A^{(t)}(\rho) = $ DYNAMICMINCUT$(G_t, \rho_t, 1/\sqrt{T})$. For the quadratic objective use Algorithm 4 (Appendix C.5.2) to set $A^{(t)}(\rho) = $ HARMONICFEEDBACKSET$(G_t, \rho_t, 1/\sqrt{T})$.
8:     Compute average loss function $l_t(\rho) = \frac{1}{|U_t|} \sum_{u \in U} l(A_{G_t(\rho), L_t, U_t}(u), \tau(u))$.
9:     For each $\rho \in C$, set $w_{t+1}(\rho) = e^{\lambda \hat{l}_t(\rho)} w_t(\rho)$, where $\hat{l}_t(\rho) = \frac{\mathbb{I}[\rho \in A^{(t)}(\rho)]}{\int_{A^{(t)}(\rho)} p_t(\rho)} l_t(\rho)$.

---

For the case of learning the unweighted threshold graph, computing the feedback set containing a given $r$ is easy as we only need the next and previous thresholds from among the $O(n^2)$ values of pairwise distances where loss may be discontinuous in $r$. We present algorithms for computing the semi-bandit feedback sets (constant performance interval containing any $\sigma$) for the weighted graph setting (Definition 1c). We propose a novel hybrid combinatorial-continuous algorithm for the mincut objective (Algorithm 3, Appendix C.5.1) which re-computes the mincut in a graph with dynamic edge weights by flow decomposition and careful flow augmentation as $\sigma$ is varied until a new mincut

is detected. For the harmonic objective, we can obtain similar efficiency (Algorithm 4, Appendix C.5.2). We seek points where $f_u(\sigma) = \frac{1}{2}$ for some $u \in U$ closest to given $\sigma_0$. For each $u$ we can find the local minima of $\left(f_u(\sigma) - \frac{1}{2}\right)^2$ or simply the root of $f_u(\sigma) - \frac{1}{2}$ using gradient descent or Newton's method. The gradient computation uses matrix inversion which can be computed in $O(n^3)$ time, and we can obtain quadratic convergence rates for finding the root. Formally, we establish Theorem 9 (Appendix C.5).

**Theorem 9.** *For the each objective in Table 1 and exponential kernel (Definition 1c), there exists an algorithm which outputs the interval containing $\sigma$ in time $\tilde{O}(n^4)$.*

## 5 Distributional setting

In the distributional setting, we are presented with instances of the problem assumed to be drawn from an unknown distribution $\mathcal{D}$ and want to learn the best value of the graph parameter $\rho$, that is one that minimizes loss $l_{A(G(\rho),L,U)}$, in expectation over the data distribution $\mathcal{D}$. We show a divergence in the weighted and unweighted graph learning problems. We analyze and provide asymptotically tight bounds for the pseudodimension of the set of loss functions parameterized by the graph family parameter $\rho$, i.e. $\mathcal{H}_\rho = \{l_{A(G(\rho),L,U)} \mid \rho \in \mathcal{P}\}$. For learning the unweighted threshold graphs, the pseudodimension is $O(\log n)$ which implies existence of an efficient algorithm with generalization guarantees in this setting. However, the pseudodimension is shown to be $\Omega(n)$ for the weighted graph setting, and therefore smoothness assumptions are necessary for learning over the algorithm family. Both these bounds are shown to be tight up to constant factors.

We also establish uniform convergence guarantees. For the unweighted graph setting, our pseudodimension bounds are sufficient for uniform convergence. We resort to bounding the Rademacher complexity in the weighted graph setting which allows us to prove distribution dependent generalization guarantees, that hold under distributional niceness assumptions of Section 4.1 (unlike pseudodimension which gives generalization guarantees that are worst-case over the distribution). The online learning results above only work for smoothed but adversarial instances, while the pseudodimension-based distributional learning sample complexity results work for any type (no smoothness needed) of independent and identically distributed instances. So these results are not superseded by the online learning results and provide new upper and lower bounds for the problem.

**Pseudodimension bounds.** We provide an upper bound on the pseudodimension of the set of loss functions for unweighted graphs $\mathcal{H}_r = \{l_{A(G(r),L,U)} \mid 0 \leq r < \infty\}$, where $G(r)$ is specified by Definition 1a. Our bounds hold for general quadratic objectives (Table 1) and imply learnability with polynomially many samples. For the upper bound, we show that given any $m$ instances we can partition the real line into $O(mn^2)$ intervals such that all values of $r$ behave identically for all instances within any fixed interval. We also show an asymptotically tight lower bound on the pseudodimension of $\mathcal{H}_r$, by presenting a collection of graph thresholds and precisely designed labeling instances which are shattered by the thresholds. For full proof details see Appendix D.

**Theorem 10.** *The pseudo-dimension of $\mathcal{H}_r$ is $\Theta(\log n)$, where $n$ is number of graph nodes.*

*Proof Sketch. Upper bound.* As $r$ is increased from 0 to infinity, at most $\binom{n}{2} + 1$ distinct graphs may be obtained. Thus given set $\mathcal{S}$ of $m$ instances $(A^{(i)}, L^{(i)})$, we can partition the real line into $O(mn^2)$ intervals such that all values of $r$ behave identically for all instances within any fixed interval. The loss function is a piecewise constant with only $O(mn^2)$ pieces. Each piece can have a witness above or below it as $r$ is varied for the corresponding interval, and so the binary labeling of $\mathcal{S}$ is fixed in that interval. The pseudo-dimension $m$ satisfies $2^m \leq O(mn^2)$ and is therefore $O(\log n)$.
*Lower bound*: We have three labeled nodes, $a_1$ with label 0 and $b_1, b_2$ labeled 1, and $n' = O(n)$ unlabeled nodes $U = \{u_1, \ldots, u_{n'}\}$. We can show that given a sequence $\{r_1, \ldots, r_{n'}\}$ of values of $r$, it is possible to construct an instance with suitable true labels of $U$ such that the loss as a function of $r$ oscillates above and below some witness as $r$ moves along the sequence of intervals $(r_i, r_{i+1})_{i \geq 0}$. At the initial threshold $r_0$, all unlabeled points have a single incident edge, connecting to $a_1$, so all predicted labels are 0. As the threshold is increased to $r_i$, (the distances are set so that) $u_i$ gets connected to both nodes with label 1 and its predicted label changes to 1. If the sequence of nodes $u_i$ is alternately labeled, the loss decreases and increases alternately as all the predicted labels turn to 1 as $r$ is increased to $r_{n'}$. This oscillation between a high and a low value can be achieved for any

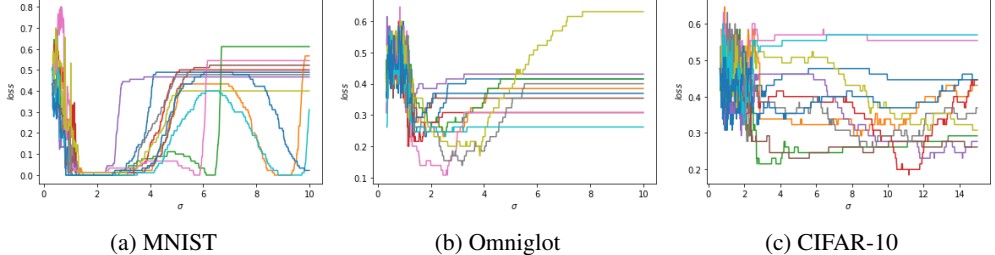

|              |              |              |
|:------------:|:------------:|:------------:|
| (a) MNIST | (b) Omniglot | (c) CIFAR-10 |

Figure 2: Multiple instances of the same problem, loss as a function of $\sigma$.

subsequence of distances $r_1, \ldots, r_{n'}$, and a witness may be set as a loss value between the oscillation limits. By precisely choosing the subsequences so that the oscillations align with the bit flips in the binary digit sequence, we can construct $m$ instances which satisfy the $2^m$ shattering constraints. $\qquad\square$

For learning weighted graphs $G(\sigma)$, we can show a $\Theta(n)$ bound on the pseudodimension of the set of loss functions $\mathcal{H}_\sigma = \{l_{A(G(\sigma),L,U)} \mid 0 \le \sigma < \infty\}$. The lower bound consists of inductively constructed graphs with carefully set edges in a precisely designed sequence (Appendix D).

**Theorem 11.** *The pseudo-dimension of $\mathcal{H}_\sigma$ is $\Theta(n)$.*

**Uniform convergence.** Our results above implies a uniform convergence guarantee for the offline distributional setting, for both weighted and unweighted graph families. For the unweighted case, we can use the pseudodimension bounds above, and for the weighted case we use dispersion guarantees from section 4.1. For either case it suffices to bound the empirical Rademacher complexity. We will need the following theorem (slightly rephrased) from Balcan et al. [2018b].

**Theorem 12.** *[Balcan et al., 2018b] Let $\mathcal{F} = \{f_\rho : \mathcal{X} \to [0,1], \rho \in \mathcal{C} \subset \mathbb{R}^d\}$ be a parametereized family of functions, where $\mathcal{C}$ lies in a ball of radius $R$. For any set $\mathcal{S} = \{x_i, \ldots, x_T\} \subseteq \mathcal{X}$, suppose the functions $u_{x_i}(\rho) = f_\rho(x_i)$ for $i \in [T]$ are piecewise L-Lipschitz and $\beta$-dispersed. Then $\hat{R}(\mathcal{F}, \mathcal{S}) \le O(\min\{\sqrt{(d/T)\log RT} + LT^{-\beta}, \sqrt{Pdim(\mathcal{F})/T}\})$.*

Now, using classic results from learning theory, we conclude that ERM has good generalization.

**Theorem 13.** *For both weighted and unweighted graph $w(u,v)$ defined above, with probability at least $1 - \delta$, the average loss on any sample $x_1, \ldots, x_T \sim D^T$, the loss suffered w.r.t. to any parameter $\rho \in \mathbb{R}^d$ satisfies $|\frac{1}{T}\sum_{i=1}^{T} l_\rho(x_i) - \mathbb{E}_{x \sim D} l_\rho(x)| \le O\left(\sqrt{\frac{d\log T \log 1/\delta}{T}}\right)$.*

## 6 Experiments

In this section we evaluate the performance of our learning procedures when finding application-specific semi-supervised learning algorithms (i.e. graph parameters). Our experiments[3] demonstrate that the best parameter for different applications varies greatly, and that the techniques presented in this paper can lead to large gains. We look at image classification based on standard pixel embedding.

*Setup*: We consider the task of semi-supervised classfication on image datasets. We restrict our attention to binary classification and pick two classes (labels 0 or 1) for each dataset. We then draw random subsets of the dataset (with class restriction) of size $n = 100$ and randomly select $L$ examples for labeling. For any data subset $S$, we measure distance between any pairs of images using the $L_2$ distance between their pixel intensities. We would like to determine data-specific good values for $\sigma$, when predictions are made by optimizing the harmonic objective (Table 1). We use three popular benchmark datasets — MNIST [LeCun et al., 1998], Omniglot [Lake et al., 2015] and CIFAR-10 [Szegedy et al., 2015]. We generate a random semi-supervised learning instance from the data by sampling 100 random examples and further sampling $L$ random examples from the subset for labeling. $L = 10$ for MNIST, while $L = 20$ for Omniglot and CIFAR-10.

---

[3]Code: `https://drive.google.com/drive/folders/1IqIw2Mp23W35UUwlz1hy24Eba5sPpVH_`

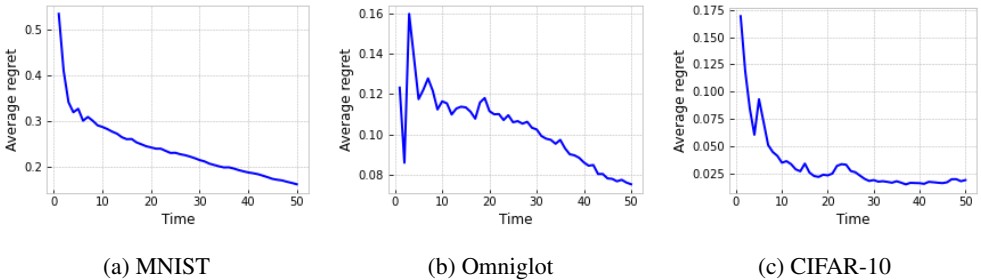

|(a) MNIST|(b) Omniglot|(c) CIFAR-10|

Figure 3: Average regret vs. $T$ for online learning of parameter $\sigma$

*Results and discussion*: For the MNIST dataset we get optimal parameters with near-perfect classification even with small values of $L$, while for other datasets the error of the optimal parameter is over $0.1$ even with larger values of $L$, indicating differences in the inherent difficulties of the classification tasks (like label noise and how well separated the classes are). We examine the full variation of performance of graph-based semi-supervised learning for all possible graphs $G(\sigma)$ for $\sigma \in [0, 10]$. The losses are piecewise constant and can have large discontinuities in some cases. The optimal parameter values vary with the dataset, but we observe at least 10%, and up to 80%, absolute gaps in performance between optimal and suboptimal values within the same dataset.

Another interesting observation is the variation of optima across data subsets, indicating transductively optimal parameters may not generalize well. We plot the variation of loss with parameter $\sigma$ for several subsets of the same size $N = 100$ for MNIST and Omniglot datasets in Figure 2. In MNIST we have two optimal ranges in most subsets but only one shared optimum (around $\sigma = 2$) across different subsets. This indicates that local search based techniques that estimate the optimal parameter values on a given data instance may lead to very poor performance on unseen instances. The CIFAR-10 example further shows that the optimal algorithm may not be easy to empirically discern.

We also implement our online algorithms and compute the average regret for finding the optimal graph parameter $\sigma$ for the different datasets. To obtain smooth curves we plot the average over 50 iterations for learning from 50 problem instances each ($T = 50$, Figure 3). We observe fast convergence to the optimal parameter regret for all the datasets considered. The starting part of these curves ($T = 0$) indicates regret for randomly setting the graph parameters, averaged over iterations, which is strongly outperformed by our learning algorithms as they learn from problem instances.

## 7 Ethics and broader impact

This work takes a step in making semi-supervised learning techniques domain independent and more practically effective. The resulting automation reduces dependence on human labelers and domain experts needed in current approaches. Dataset bias and ethics of applications will need to be individually considered when applying our approach to real world problems.

## 8 Acknowledgments

This material is based on work supported by the National Science Foundation under grants CCF-1535967, CCF-1910321, IIS-1618714, IIS-1901403, and SES-1919453; the Defense Advanced Research Projects Agency under cooperative agreement HR00112020003; an AWS Machine Learning Research Award; an Amazon Research Award; a Bloomberg Research Grant; a Microsoft Research Faculty Fellowship. The views expressed in this work do not necessarily reflect the position or the policy of the Government and no official endorsement should be inferred.

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
