## Appendix

## A Definitions from previous work

We first note the definitions of some well-known useful learning theoretic complexity measures. Recall the definitions of pseudodimension and Rademacher complexity, well-known measures for hypothesis-space complexity in statistical learning theory. Bounding these quantities implies immediate bounds on learning error using classic learning theoretic results. In Section 5 we bound the pseudodimension and Rademacher complexity for the problems of learning unweighted and weighted graphs.

**Definition 14.** *Pseudo-dimension [Pollard, 2012]. Let $\mathcal{H}$ be a set of real valued functions from input space $\mathcal{X}$. We say that $C = (x_1, \ldots, x_m) \in \mathcal{X}^m$ is pseudo-shattered by $\mathcal{H}$ if there exists a vector $r = (r_1, \ldots, r_m) \in \mathbb{R}^m$ (called "witness") such that for all $b = (b_1, \ldots, b_m) \in \{\pm 1\}^m$ there exists $h_b \in \mathcal{H}$ such that $sign(h_b(x_i) - r_i) = b_i$. Pseudo-dimension of $\mathcal{H}$ is the cardinality of the largest set pseudo-shattered by $\mathcal{H}$.*

**Definition 15.** *Rademacher complexity [Bartlett and Mendelson, 2002]. Let $\mathcal{F} = \{f_\rho : \mathcal{X} \to [0,1], \rho \in \mathcal{C} \subset \mathbb{R}^d\}$ be a parameterized family of functions, and sample $\mathcal{S} = \{x_i, \ldots, x_T\} \subseteq \mathcal{X}$. The empirical Rademacher complexity of $\mathcal{F}$ with respect to $\mathcal{S}$ is defined as $\hat{R}(\mathcal{F}, \mathcal{S}) = \mathbb{E}_\sigma \left[ \sup_{f \in \mathcal{F}} \frac{1}{T} \sum_{i=1}^T \sigma_i f(x_i) \right]$, where $\sigma_i \sim U(\{-1, 1\})$ are Rademacher variables.*

We will also need the definition of *dispersion* which, informally speaking, captures how amenable a non-Lipschitz function is to online learning. As noted in [Balcan et al., 2018b, 2020c], dispersion is necessary and sufficient for learning piecewise Lipschitz functions.

**Definition 16.** *Dispersion [Balcan et al., 2020b]. The sequence of random loss functions $l_1, \ldots, l_T$ is $\beta$-dispersed for the Lipschitz constant $L$ if, for all $T$ and for all $\epsilon \geq T^{-\beta}$, we have that, in expectation, at most $\tilde{O}(\epsilon T)$ functions (the soft-O notation suppresses dependence on quantities beside $\epsilon, T$ and $\beta$, as well as logarithmic terms) are not $L$-Lipschitz for any pair of points at distance $\epsilon$ in the domain $\mathcal{C}$. That is, for all $T$ and for all $\epsilon \geq T^{-\beta}$,*

$$\mathbb{E} \left[ \max_{\substack{\rho, \rho' \in \mathcal{C} \\ \|\rho - \rho'\|_2 \leq \epsilon}} \left| \{t \in [T] \mid l_t(\rho) - l_t(\rho') > L \|\rho - \rho'\|_2 \} \right| \right] \leq \tilde{O}(\epsilon T).$$

## B Motivation for data-driven design

**Theorem 5.** *Let $r_{\min}$ denote the smallest value of threshold $r$ for which every unlabeled node of $G(r)$ is reachable from some labeled node, and $r_{\max}$ be the smallest value of threshold $r$ for which $G(r)$ is the complete graph. There exists a data instance $(L, U)$ such that for any $r_\zeta = \zeta r_{\min} + (1 - \zeta) r_{\max}$ for $\zeta \in (0, 1)$, there exists a set of labelings $\mathcal{U}$ of the unlabeled points such that for some $U_\zeta, \bar{U}_\zeta \in \mathcal{U}$, $r_\zeta$ minimizes $l_{A(G(r), L, U_\zeta)}$ but not $l_{A(G(r), L, \bar{U}_\zeta)}$.*

*Proof.* Note that for any $r < r_{\min}$, there is no graph similarity information for at least one node, and therefore all labels cannot be predicted. Also, the graph is unchanged for all $r \geq r_{\max}$. Therefore, $r \in [r_{\min}, r_{\max}]$ captures all graphs of interest on a given data instance.

Intuitively the statement claims that any threshold $r$ (modulo the scaling factors for the data embedding) may be optimal or suboptimal for some data labeling for a given constructed instance. Therefore it is useful to consider several problem instances and learn the optimal value of $r$ for the data distribution. We will present an example where an unlabeled point is closest to some labeled point of one class but closer to more points of another class on average. So for small thresholds it may be labeled as the first class and for larger thresholds as the second class.

Let $L = L_1 \cup L_2$ with $|L_1| < |L_2|$ and $d(u, v) = 0$ for $u, v \in L_i, i \in \{1, 2\}$, $d(u, v) = 3r^*/2$ for $u \in L_i, v \in L_j, i \neq j$, where $r^*$ is a positive real. Further let $U = \{a\}$ such that $d(a, u_i) = ir^*/2$ for each $u_i \in L_i$. It is straightforward to verify that the triangle inequality is satisfied. Further note that $r_{\min} = r^*/2$ and $r_{\max} = 3r^*/2$. Our set of labelings $\mathcal{U}$ will include one that labels $a$ according to each class. Now we have two cases

1. $\zeta \in (0, \frac{1}{2})$: $r_{\min} \leq r < r^*$, $G(r_\zeta)$ connects $a$ to $L_1$ but not $L_2$ and we have that the loss is minimized exactly for the labeling where $a$ matches $L_1$.

2. $\zeta \in [\frac{1}{2}, 1)$: $r^* \leq r \leq r_{\max}$, $G(r_\zeta)$ connects $a$ to both $L_1$ and $L_2$. But since $|L_1| < |L_2|$, we predict that the label of $a$ matches that of $L_2$.

Finally we note that $d(u, v)$ may not be exactly zero when $u \neq v$ for a metric. This is easily fixed by making tiny perturbations to the labeled points, for any given $r_\zeta$. $\qquad\square$

The example presented above captures some essential challenges of our setting in the following sense. Firstly, we see that the loss function may be non-Lipschitz (as a function of the parameter $r$), which makes the optimization problem more challenging. More importantly, it highlights that graph similarity only approximately corresponds to label similarity, and how the accuracy of this correspondence is strongly influenced by the graph parameters. In this sense, it may not be possible to learn from a single instance, and considering a data-driven setting is crucial.

## C Dispersion and Online learning

In this appendix we include details of proofs and algorithms from section 4.1.

### C.1 Dispersion for threshold graphs

We will need the following simple lemma.

**Lemma 17.** *Let* $\bar{l}(r) = l_{A(G(r), L, U)}$ *be the loss function for graph* $G(r)$ *created using the threshold kernel* $w(u, v) = \mathbb{I}[d(u, v) \leq r]$. *Then* $\bar{l}(r)$ *is piecewise constant and any discontinuity occurs at* $r^* = d(u, v)$ *for some graph nodes* $u, v$.

*Proof.* This essentially follows from the observation that as $r$ is increased, the graph gets a new edge only for some $r^* = d(u, v)$. Therefore no matter what the optimization algorithm is used to predict labels to minimize the loss, the loss is fixed given the graph, and has discontinuities potentially only when new edges are added. $\qquad\square$

We are now ready to establish dispersion for the unweighted graph setting.

**Theorem 6.** *Let* $l_1, \ldots, l_T : \mathbb{R} \to \mathbb{R}$ *denote an independent sequence of losses as a function of parameter* $r$, *when the graph is created using a threshold kernel* $w(u, v) = \mathbb{I}[d(u, v) \leq r]$ *and labeled by applying any algorithm on the graph. If* $d(u, v)$ *follows a* $\kappa$-*bounded distribution for any* $u, v$, *the sequence is* $\frac{1}{2}$-*dispersed, and the regret of Algorithm 1 is* $\tilde{O}(\sqrt{T})$.

*Proof.* Assume a fixed but arbitrary ordering of nodes in each $V_t = L_t \cup U_t$ denoted by $V_t^{(i)}, i \in [n]$. Define $d_{i,j} = \{d(u, v) \mid u = V_t^{(i)}, v = V_t^{(j)}, t \in [T]\}$. Since $d_{i,j}$ is $\kappa$-bounded, the probability that it falls in any interval of length $\epsilon$ is $O(\kappa\epsilon)$. Since different problem instances are independent and using the fact that the VC dimension of intervals is 2, with probability at least $1 - \delta/D$, every interval of width $\epsilon$ contains at most $O\left(\kappa\epsilon T + \sqrt{T \log D/\delta}\right)$ discontinuities from each $d_{i,j}$ (using Lemma 17). Now a union bound over the failure modes for $d_{i,j}$ for different $i, j$ gives $O\left(n^2 \kappa\epsilon T + n^2 \sqrt{T \log n/\delta}\right)$ discontinuities with probability at least $1 - \delta$ for any $\epsilon$-interval. Setting $\delta = 1/\sqrt{T}$, for each $\epsilon \geq 1/\sqrt{T}$ the maximum number of discontinuities in any $\epsilon$-interval is at most $(1 - \delta)O\left(\kappa n^2 \sqrt{T \log n\sqrt{T}}\right) + \delta T = \tilde{O}(\epsilon T)$, in expectation, proving $\frac{1}{2}$-dispersion. $\qquad\square$

### C.2 A general tool for analyzing dispersion

If the weights of the graph are given by a polynomial kernel $w(u, v) = (\tilde{d}(u, v) + \tilde{\alpha})^d$, we can apply the general tool developed by Balcan et al. [2020b] to learn $\tilde{\alpha}$, which we summarize below.

1. Bound the probability density of the random set of discontinuities of the loss functions.

2. Use a VC-dimension based uniform convergence argument to transform this into a bound on the dispersion of the loss functions.

Formally, we have the following theorems from Balcan et al. [2020b], which show how to use this technique when the discontinuities are roots of a random polynomial.

**Theorem 18** (Balcan et al. [2020b]). *Consider a random degree $d$ polynomial $\phi$ with leading coefficient 1 and subsequent coefficients which are real of absolute value at most $R$, whose joint density is at most $\kappa$. There is an absolute constant $K$ depending only on $d$ and $R$ such that every interval $I$ of length $\leq \epsilon$ satisfies $Pr(\phi$ has a root in $I) \leq \kappa\epsilon/K$.*

**Theorem 19** (Balcan et al. [2020b]). *Let $l_1, \ldots, l_T : \mathbb{R} \to \mathbb{R}$ be independent piecewise $L$-Lipschitz functions, each having at most $K$ discontinuities. Let $D(T, \epsilon, \rho) = |\{1 \leq t \leq T \mid l_t$ is not $L$-Lipschitz on $[\rho - \epsilon, \rho + \epsilon]\}|$ be the number of functions that are not $L$-Lipschitz on the ball $[\rho - \epsilon, \rho + \epsilon]$. Then we have $E[\max_{\rho \in \mathbb{R}} D(T, \epsilon, \rho)] \leq \max_{\rho \in \mathbb{R}} E[D(T, \epsilon, \rho)] + O(\sqrt{T \log(TK)})$.*

We will now use Theorems 18 and 19 to establish dispersion in our setting. We first need a simple lemma about $\kappa$-bounded distributions. We remark that similar properties have been proved in Balcan et al. [2018b, 2020b], in other problem contexts. Specifically, Balcan et al. [2018b] show the lemma for a ratio of random variables, $Z = X/Y$, and Balcan et al. [2020b] establish it for the sum $Z = X + Y$ but for independent variables $X, Y$.

**Lemma 20.** *Suppose $X$ and $Y$ are real-valued random variables taking values in $[m, m + M]$ and $[m', m' + M']$ for some $m, m', M, M' \in \mathbb{R}^+$ and suppose that their joint distribution is $\kappa$-bounded. Then,*

(i) *$Z = X + Y$ is drawn from a $K_1\kappa$-bounded distribution, where $K_1 \leq \min\{M, M'\}$.*

(ii) *$Z = XY$ is drawn from a $K_2\kappa$-bounded distribution, where $K_2 \leq \min\{M/m, M'/m'\}$.*

*Proof.* Let $f_{X,Y}(x, y)$ denote the joint density of $X, Y$.

(i) The case where $X, Y$ are independent has been studied (Lemma 25 in Balcan et al. [2020b]), the following is slightly more involved. The cumulative density function for $Z$ is given by

$$F_Z(z) = \Pr(Z \leq z) = \Pr(X + Y \leq z) = \Pr(X \leq z - Y)$$
$$= \int_{m'}^{m'+M'} \int_m^{z-y} f_{X,Y}(x, y) dx dy.$$

The density function for $Z$ can be obtained using Leibniz's rule as

$$f_Z(z) = \frac{d}{dz} F_Z(z) = \frac{d}{dz} \int_{m'}^{m'+M'} \int_m^{z-y} f_{X,Y}(x, y) dx dy$$
$$= \int_{m'}^{m'+M'} \left( \frac{d}{dz} \int_m^{z-y} f_{X,Y}(x, y) dx \right) dy$$
$$= \int_{m'}^{m'+M'} f_{X,Y}(z - y, y) dy$$
$$\leq M'\kappa.$$

A symmetric argument shows that $f_Z(z) \leq M\kappa$, together with above this completes the proof.

(ii) The cumulative density function for $Z$ is given by

$$F_Z(z) = \Pr(Z \leq z) = \Pr(XY \leq z) = \Pr(X \leq z/Y)$$
$$= \int_{m'}^{m'+M'} \int_m^{z/y} f_{X,Y}(x, y) dx dy.$$

The density function for $Z$ can be obtained using Leibniz's rule as

$$f_Z(z) = \frac{d}{dz}F_Z(z) = \frac{d}{dz}\int_{m'}^{m'+M'}\int_m^{z/y} f_{X,Y}(x,y)dxdy$$

$$= \int_{m'}^{m'+M'}\left(\frac{d}{dz}\int_m^{z/y} f_{X,Y}(x,y)dx\right)dy$$

$$= \int_{m'}^{m'+M'}\frac{1}{y}f_{X,Y}(z/y,y)dy$$

$$\leq \int_{m'}^{m'+M'}\frac{1}{m'}f_{X,Y}(z/y,y)dy$$

$$\leq \frac{M'}{m'}\kappa.$$

Similarly we can show that $f_Z(z) \leq M\kappa/m$, together with above this completes the proof.

$\square$

**Theorem 7.** *Let $l_1,\ldots,l_T : \mathbb{R} \to \mathbb{R}$ denote an independent sequence of losses as a function of $\tilde{\alpha}$, for graph with edges $w(u,v) = (\tilde{d}(u,v) + \tilde{\alpha})^d$ labeled by optimizing the quadratic objective $\sum_{u,v} w(u,v)(f(u)-f(v))^2$. If $\tilde{d}(u,v)$ follows a $\kappa$-bounded distribution with a closed and bounded support, the sequence is $\frac{1}{2}$-dispersed, and the regret of Algorithm 1 may be upper bounded by $\tilde{O}(\sqrt{T})$.*

*Proof.* $w(u,v)$ is a polynomial in $\tilde{\alpha}$ of degree $d$ with coefficient of $\tilde{\alpha}^i$ given by $c_i = D_{d,i}\tilde{d}(u,v)^{E_{d,i}}$ for $i \in [d]$. Since the support of $\tilde{d}(u,v)$ is closed and bounded, we have $m \leq \tilde{d}(u,v) \leq M$ with probability 1 for some $M > 1, m > 0$ (since $\tilde{d}(u,v)$ is a metric, $\tilde{d}(u,v) > 0$ for $u \neq v$).

To apply Theorem 18, we note that we have an upper bound on the coefficients, $R < (dM)^d$. Moreover, if $f(x)$ denotes the probability density of $d(u,v)$ and $F(x)$ its cumulative density,

$$\Pr(c_i \leq x_i) = \Pr\left(D_{d,i}\tilde{d}(u,v)^{E_{d,i}} \leq x_i\right)$$

$$= \Pr\left(\tilde{d}(u,v) \leq \left(\frac{x_i}{D_{d,i}}\right)^{1/E_{d,i}}\right) = F\left(\left(\frac{x_i}{D_{d,i}}\right)^{1/E_{d,i}}\right).$$

Thus,

$$\Pr(c_i \leq x_i \text{ for each } i \in [d]) = F\left(\min_i\left(\frac{x_i}{D_{d,i}}\right)^{1/E_{d,i}}\right).$$

The joint density of the coefficients is therefore $K\kappa$-bounded where $K$ only depends on $d, m$. ($K \leq \max_i D_{d,i}^{-1/E_{d,i}}m^{-1+1/E_{d,i}}$).

Consider the harmonic solution of the quadratic objective Zhu et al. [2003] which is given by $f_U = (D_{UU} - W_{UU})^{-1}W_{UL}f_L$. For any $u \in U$, $f(u) = 1/2$ is a polynomial equation in $\tilde{\alpha}$ with degree at most $nd$. The coefficients of these polynomials are formed by multiplying sets of weights $w(u,v)$ of size up to $n$ and adding the products, and are also bounded density on a bounded support (using above observation in conjunction with Lemma 20). The dispersion result now follows by an application of Theorems 18 and 19. The regret bound is implied by results from Balcan et al. [2018b, 2020c]. $\square$

### C.3  Dispersion for roots of exponential polynomials

In this section we will extend the applicability of the dispersion analysis technique from Appendix C.2 to exponential polynomials, i.e. functions of the form $\phi(x) = \sum_{i=1}^n a_i e^{b_i x}$. We will now extend the analysis to obtain similar results when using the exponential kernel $w(u,v) = e^{-||u-v||^2/\sigma^2}$. The results of Balcan et al. [2020b] no longer directly apply as the points of discontinuity are no longer roots of polynomials. To this end, we extend and generalize arguments from Balcan et al. [2020b] below. We need to generalize Theorem 18 to exponential polynomials below.

**Theorem 21.** *Let $\phi(x) = \sum_{i=1}^{n} a_i e^{b_i x}$ be a random function, such that coefficients $a_i$ are real and of magnitude at most $R$, and distributed with joint density at most $\kappa$. Then for any interval $I$ of width at most $\epsilon$, $P(\phi$ has a zero in $I) \leq \tilde{O}(\epsilon)$ (dependence on $b_i, n, \kappa, R$ suppressed).*

*Proof.* For $n = 1$ there are no roots, so assume $n > 1$. Suppose $\rho$ is a root of $\phi(x)$. Then $\mathbf{a} = (a_1, \ldots, a_n)$ is orthogonal to $\varrho(\rho) = (e^{b_1 \rho}, \ldots, e^{b_n \rho})$ in $\mathbb{R}^n$. For a fixed $\rho$, the set $S_\rho$ of coefficients $\mathbf{a}$ for which $\rho$ is a root of $\phi(y)$ lie along an $n - 1$ dimensional linear subspace of $\mathbb{R}^n$. Now $\phi$ has a root in any interval $I$ of length $\epsilon$, exactly when the coefficients lie on $S_\rho$ for some $\rho \in I$. The desired probability is therefore upper bounded by $\max_\rho \text{VOL}(\cup S_y \mid y \in [\rho - \epsilon, \rho + \epsilon]) / \text{VOL}(S_y \mid y \in \mathbb{R})$ which we will show to be $\tilde{O}(\epsilon)$. The key idea is that if $|\rho - \rho'| < \epsilon$, then $\varrho(\rho)$ and $\varrho(\rho')$ are within a small angle $\theta_{\rho, \rho'} = \tilde{O}(\epsilon)$ for small $\epsilon$ (the probability bound is vacuous for large $\epsilon$). But any point in $S_\rho$ is at most $\tilde{O}(\theta_{\rho, \rho'})$ from a point in $S_{\rho'}$, which implies the desired bound (similar arguments to Theorem 18).

We will now flesh out the above sketch. Indeed,

$$\sin \theta_{\rho, \rho'} = \sqrt{1 - \frac{(\langle \varrho(\rho), \varrho(\rho') \rangle)^2}{\|\varrho(\rho)\| \|\varrho(\rho')\|}}$$

$$= \sqrt{1 - \frac{(\sum_i e^{b_i \rho} e^{b_i \rho'})^2}{\sum_i e^{2b_i \rho} \sum_i e^{2b_i \rho'}}}$$

$$= \sqrt{\frac{\sum_{i \neq j} e^{2(b_i \rho + b_j \rho')} - e^{(b_i + b_j)(\rho + \rho')}}{\sum_i e^{2b_i \rho} \sum_i e^{2b_i \rho'}}}.$$

Now, for $\rho' = \rho + \varepsilon$, $|\varepsilon| < \epsilon$,

$$\sin \theta_{\rho, \rho'} = \sqrt{\frac{\sum_{i \neq j} e^{2(b_i \rho + b_j \rho + b_j \varepsilon)} - e^{(b_i + b_j)(2\rho + \varepsilon)}}{\sum_i e^{2b_i \rho} \sum_i e^{2b_i \rho'}}}$$

$$= \sqrt{\frac{\sum_{i \neq j} e^{2\rho(b_i + b_j)}(e^{2b_j \varepsilon} - e^{(b_i + b_j)\varepsilon})}{\sum_i e^{2b_i \rho} \sum_i e^{2b_i \rho'}}}.$$

Using the Taylor's series approximation for $e^{2b_j \varepsilon}$ and $e^{(b_i + b_j)\varepsilon}$, we note that the largest terms that survive are quadratic in $\varepsilon$. $\sin \theta_{\rho, \rho'}$, and therefore also $\theta_{\rho, \rho'}$, is $\tilde{O}(\epsilon)$.

Next it is easy to show that any point in $S_\rho$ is at most $\tilde{O}(\theta_{\rho, \rho'})$ from a point in $S_{\rho'}$. For $n = 2$, $S_\rho$ and $S_{\rho'}$ are along lines orthogonal to $\rho$ and $\rho'$ and are thus themselves at an angle $\theta_{\rho, \rho'}$. Since we further assume that the coefficients are bounded by $R$, any point on $S_\rho$ is within $O(R\theta_{\rho, \rho'}) = \tilde{O}(\theta_{\rho, \rho'})$ of the nearest point in $S_{\rho'}$. For $n > 2$, consider the 3-space spanned by $\rho, \rho'$ and an arbitrary $\varsigma \in S_\rho$. $S_\rho$ and $S_{\rho'}$ are along 2-planes in this space with normal vectors $\rho, \rho'$ respectively. Again it is straightforward to see that the nearest point in the projection of $S_{\rho'}$ to $\varsigma$ is $\tilde{O}(\theta_{\rho, \rho'})$.

The remaining proof is identical to that of Theorem 18 (see Theorem 18 of Balcan et al. [2020b]), and is omitted for brevity.

$\square$

We will also need the following lemma for the second step noted above, i.e. obtain a result similar to Theorem 19 for exponential polynomials.

**Lemma 22.** *The equation $\sum_{i=1}^{n} a_i e^{b_i x} = 0$ where $a_i, b_i \in \mathbb{R}$ has at most $n - 1$ distinct solutions $x \in \mathbb{R}$.*

*Proof.* We will use induction on $n$. It is easy to verify that there is no solution for $n = 1$. We assume the statement holds for all $1 \leq n \leq N$. Consider the equation $\phi_{N+1}(x) = \sum_{i=1}^{N+1} a_i e^{b_i x} = 0$.

WLOG $a_1 \neq 0$ and we can write

$$\phi_{N+1}(x) = \sum_{i=1}^{N+1} a_i e^{b_i x} = a_1 e^{b_1 x} \left( 1 + \sum_{i=2}^{N+1} \frac{a_i}{a_1} e^{(b_i - b_1)x} \right) = a_1 e^{b_1 x} \left( 1 + g(x) \right).$$

By our induction hypothesis, $g'(0) = 0$ has at most $N - 1$ solutions, and so $(1 + g(x))'$ has at most $N - 1$ roots. By Rolle's theorem, $(1 + g(x))$ has at most $N$ roots, and therefore $\phi_{N+1}(x) = 0$ has at most $N$ solutions. □

Lemma 22 implies that Theorem 19 may be applied. The number of discontinuities may be exponentially high in this case. Indeed solving the quadratic objective can result in an exponential equation of the form in Lemma 22 with $O(|U|^n)$ terms.

### C.4 Learning several metrics simultaneously

We start by getting a couple useful definitions out of the way.

**Definition 23** (Homogeneous algebraic hypersurface). *An algebraic hypersurface is an algebraic variety (a system of polynomial equations) that may be defined by a single implicit equation of the form $p(x_1, \ldots, x_n) = 0$, where $p$ is a multivariate polynomial. The degree $d$ of the algebraic hypersurface is the total degree of the polynomial $p$. We say that the algebraic hypersurface is homogeneous if $p$ is a homogeneous polynomial, i.e. $p(\lambda x_1, \ldots, \lambda x_m) = \lambda^d p(x_1, \ldots, x_n)$.*

In the following we will refer to homogeneous algebraic hypersurfaces as simply algebraic hypersurfaces. We will also need the standard definition of set shattering, which we restate in our context as follows.

**Definition 24** (Hitting and Shattering). *Let $\mathcal{C}$ denote a set of curves in $\mathbb{R}^p$. We say that a subset of $\mathcal{C}$ is hit by a curve $s$ if the subset is exactly the set of curves in $\mathcal{C}$ which intersect the curve $s$. A collection of curves $\mathcal{S}$ shatters the set $\mathcal{C}$ if for each subset $C$ of $\mathcal{C}$, there is some element $s$ of $\mathcal{S}$ such that $s$ hits $C$.*

To extend our learning results to learning graphs built from several metrics, we will now state and prove a couple theorems involving algebraic hypersurfaces. Our results generalize significantly the techniques from Balcan et al. [2020b] by bringing in novel connections with algebraic geometry.

**Theorem 3.** *There is a constant $k$ depending only on $d$ and $p$ such that axis-aligned line segments in $\mathbb{R}^p$ cannot shatter any collection of $k$ algebraic hypersurfaces of degree at most $d$.*

*Proof.* Let $\mathcal{C}$ denote a collection of $k$ algebraic hypersurfaces of degree at most $d$ in $\mathbb{R}^p$. We say that a subset of $\mathcal{C}$ is *hit* by a line segment if the subset is exactly the set of curves in $\mathcal{C}$ which intersect the segment, and *hit* by a line if some segment of the line hits the subset. We seek to upper bound the number of subsets of $\mathcal{C}$ which may be hit by axis-aligned line segments. We will first consider shattering by line segments in a fixed axial direction $x$. We can easily extend this to axis-aligned segments by noting they may hit only $p$ times as many subsets.

Let $L_c$ be a line in the $x$ direction. The subsets of $\mathcal{C}$ which may be hit by (segments along) $L_c$ is determined by the pattern of intersections of $L_c$ with hypersurfaces in $\mathcal{C}$. By Bezout's theorem, there are at most $kd + 1$ distinct regions of $L_c$ due to the intersections. Therefore at most $\binom{kd+1}{2}$ distinct subsets may be hit.

Define the equivalence relation $L_{c_1} \sim L_{c_2}$ if the same hypersurfaces in $\mathcal{C}$ intersect $L_{c_1}$ and $L_{c_2}$, and in the same order (including with multiplicities). To determine these equivalence classes, we will project the hypersurfaces in $\mathcal{C}$ on to a hyperplane orthogonal to the $x$-direction. By the Tarski-Seidenberg-Łojasiewicz Theorem, we get a semi-algebraic collection $\mathcal{C}_x$, i.e. a set of polynomial equations and constraints in the projection space. Each cell of $\mathcal{C}_x$ corresponds to an equivalence class. Using well-known upper bounds for *cylindrical algebraic decomposition* (see for example England and Davenport [2016]), we get that the number of equivalence classes is at most $O\left( (2d)^{2^p - 1} k^{2^p - 1} 2^{2^{p-1}} \right)$.

Putting it all together, the number of subsets hit by any axis aligned segment is at most

$$O\left( p \binom{kd+1}{2} (2d)^{2^p - 1} k^{2^p - 1} 2^{2^{p-1}} \right).$$

We are done as this is less than $2^k$ for fixed $d$ and $p$ and large enough $k$, and therefore all subsets may not be hit.

$\square$

**Theorem 4.** *Let $l_1, \ldots, l_T : \mathbb{R}^p \to \mathbb{R}$ be independent piecewise L-Lipschitz functions, each having discontinuities specified by a collection of at most $K$ algebraic hypersurfaces of bounded degree. Let $L$ denote the set of axis-aligned paths between pairs of points in $\mathbb{R}^p$, and for each $s \in L$ define $D(T, s) = |\{1 \leq t \leq T \mid l_t \text{ has a discontinuity along } s\}|$. Then we have $\mathbb{E}[\sup_{s \in L} D(T, s)] \leq \sup_{s \in L} \mathbb{E}[D(T, s)] + O(\sqrt{T \log(TK)})$.*

*Proof.* The proof is similar to that of Theorem 19 (see Balcan et al. [2020b]). The main difference is that instead of relating the number of ways intervals can label vectors of discontinuity points to the VC-dimension of intervals, we instead relate the number of ways line segments can label vectors of $K$ algebraic hypersurfaces of degree $d$ to the VC-dimension of line segments (when labeling algebraic hypersurfaces), which from Theorem 3 is constant. To verify dispersion, we need a uniform-convergence bound on the number of Lipschitz failures between the worst pair of points $\alpha, \alpha'$ at distance $\leq \epsilon$, but the definition allows us to bound the worst rate of discontinuties along any path between $\alpha, \alpha'$ of our choice. We can bound the VC dimension of axis aligned segments against bounded-degree algebraic hypersurfaces, which will allow us to establish dispersion by considering piecewise axis-aligned paths between points $\alpha$ and $\alpha'$.

Let $\mathcal{C}$ denote the set of all algebraic hypersurfaces of degree $d$. For simplicity, we assume that every function has its discontinuities specified by a collection of exactly $K$ algebraic hypersurfaces. For each function $l_t$, let $\gamma^{(t)} \in \mathcal{C}^K$ denote the ordered tuple of algebraic hypersurfaces in $\mathcal{C}$ whose entries are the discontinuity locations of $l_t$. That is, $l_t$ has discontinuities along $(\gamma_1^{(t)}, \ldots, \gamma_K^{(t)})$, but is otherwise $L$-Lispchitz.

For any axis aligned path $s$, define the function $f_s : \mathcal{C}^K \to \{0, 1\}$ by

$$f_s(\gamma) = \begin{cases} 1 & \text{if for some } i \in [K] \ \gamma_i \text{ intersects } s \\ 0 & \text{otherwise,} \end{cases}$$

where $\gamma = (\gamma_1, \ldots, \gamma_K) \in \mathcal{C}^K$. The sum $\sum_{t=1}^T f_s(\gamma^{(t)})$ counts the number of vectors $(\gamma_1^{(t)}, \ldots, \gamma_K^{(t)})$ that intersect $s$ or, equivalently, the number of functions $l_1, \ldots, l_T$ that are not $L$-Lipschitz on $s$. We will apply VC-dimension uniform convergence arguments to the class $\mathcal{F} = \{f_s : \mathcal{C}^K \to \{0, 1\} \mid s \text{ is an axis-aligned path}\}$. In particular, we will show that for an independent set of vectors $(\gamma_1^{(t)}, \ldots, \gamma_K^{(t)})$, with high probability we have that $\frac{1}{T} \sum_{t=1}^T f_s(\gamma^{(t)})$ is close to $\mathbb{E}[\frac{1}{T} \sum_{t=1}^T f_s(\gamma^{(t)})]$ for all paths $s$. This uniform convergence argument will lead to the desired bounds.

Indeed, Theorem 3 implies that VC dimension of $\mathcal{F}$ is $O(\log K)$. Now standard VC-dimension uniform convergence arguments for the class $\mathcal{F}$ imply that with probability at least $1 - \delta$, for all $f_s \in \mathcal{F}$

$$\left| \frac{1}{T} \sum_{t=1}^T f_s(\gamma^{(t)}) - \mathbb{E}\left[ \frac{1}{T} \sum_{t=1}^T f_s(\gamma^{(t)}) \right] \right| \leq O\left( \sqrt{\frac{\log(K/\delta)}{T}} \right), \text{ or}$$

$$\left| \sum_{t=1}^T f_s(\gamma^{(t)}) - \mathbb{E}\left[ \sum_{t=1}^T f_s(\gamma^{(t)}) \right] \right| \leq O\left( \sqrt{T \log(K/\delta)} \right).$$

Now since $D(T, s) = \sum_{t=1}^T f_s(\gamma^{(t)})$, we have for all $s$ and $\delta$, with probability at least $1 - \delta$, $\sup_{s \in L} D(T, s) \leq \sup_{s \in L} \mathbb{E}[D(T, s)] + O(\sqrt{T \log(K/\delta)})$. Taking expectation and setting $\delta = 1/\sqrt{T}$ completes the proof as it allows us to bound the expected discontinuities by $O(\sqrt{T})$ when the above high probability event fails. $\square$

Theorem 4 above generalizes the second step of the dispersion tool from single parameter families to several hyperparameters, and uses Theorem 3 as a key ingredient. To complete the first step of in the multi-parameter setting, we can use a simple generalization of Theorem 18 by showing that few zeros are likely to occur on a piecewise axis-aligned path on whose pieces the zero sets of the multivariate polynomial is the zero set of a single-variable polynomial. Putting together we get Theorem 8.

**Theorem 8.** *Let $l_1, \ldots, l_T : \mathbb{R}^p \to \mathbb{R}$ denote an independent sequence of losses as a function of parameters $\rho_i, i \in [p]$, when the graph is created using a polynomial kernel $w(u, v) = (\sum_{i=1}^{p-1} \rho_i \tilde{d}(u, v) + \rho_p)^d$ and labeled by optimizing the quadratic objective $\sum_{u,v} w(u, v)(f(u) - f(v))^2$. If $\tilde{d}(u, v)$ follows a $\kappa$-bounded distribution with a closed and bounded support, the sequence is $\frac{1}{2}$-dispersed, and the regret of Algorithm 1 may be upper bounded by $\tilde{O}(\sqrt{T})$.*

*Proof.* Notice that $w(u, v)$ is a homogeneous polynomial in $\rho = (\rho_i, i \in [p])$. Further, the solutions of the quadratic objective subject to $f(u) = 1/2$ for some $u$ are also homogeneous polynomial equations, of degree $nd$. Now to show dispersion, consider an axis-aligned path between any two parameter vectors $\rho, \rho'$ such that $\|\rho - \rho'\| < \epsilon$ (notice that the definition of dispersion allows us to use any path between $\rho, \rho'$ for counting discontinuities). To compute the expected number of non-Lipchitzness in along this path, notice that for any fixed segment of this path, all but one variable are constant and the discontinuities are the zeros of single variable polynomial with bounded-density random coefficients, and that Theorem 18 applies. Summing along these paths we get at most $\tilde{O}(p\epsilon)$ discontinuities in expectation for any $\|\rho - \rho'\| < \epsilon$. Theorem 4 now completes the proof of dispersion in this case. $\qquad\square$

### C.5 Semi-bandit efficient algorithms

In this appendix we present details of the efficient algorithms for computing the semi-bandit feedback sets in Algorithm 2. For unweighted graphs, we only have a polynomial number $O(n^2)$ of feedback sets and the feedback set for a given $\rho_t$ is readily computed by looking up a sorted list of distances $d(u, v)_{u,v \in L_i \cup U_i}$. For the weighted graph setting, we need non-trivial algorithms.

#### C.5.1 Min-cut objective

First some notation for this section. We will use $G = (V, E)$ to denote an undirected graph with $V$ as the set of nodes and $E \subseteq V \times V$ the weighted edges with capacity $d : E \to \mathbb{R}_{\geq 0}$. We are given special nodes $s, t \in V$ called *source* and *target* vertices. Recall the following definitions.

**Definition 25. (s,t)-flows** *An (s,t)-flow (or just flow if the source and target are clear from context) is a function $f : V \times V \to \mathbb{R}_{\geq 0}$ that satisfies the conservation constraint at every vertex v except possibly s and t given by $\sum_{(u,v) \in E} f(u, v) = \sum_{(v,u) \in E} f(v, u)$. The value of flow (also refered by just flow when clear from context) is the total flow out of s, $\sum_{u \in V} f(s, u) - \sum_{u \in V} f(u, s)$.*

**Definition 26. (s,t)-cut** *An (s,t)-cut (or just a cut if the source and target are clear from context) is a partition of V into $S, T$ such that $s \in S, t \in T$. We will denote the set $\{(u, v) \in E \mid u \in S, v \in T\}$ of edges in the cut by $\partial S$ or $\partial T$. The capacity of the cut is the total capacity of edges in the cut.*

For convenience we also define

**Definition 27.** *Path flow. An (s,t)-flow is a path flow along a path $p = (s = v_0, v_1, \ldots, v_n = t)$ if $f(u, w) > 0$ iff $(u, w) = (v_i, v_{i+1})$ for some $i \in [n - 1]$.*

**Definition 28.** *Residual capacity graph. Given a set of path flows $F$, the residual capacity graph (or simply the residual graph) is the graph $G' = (V, E)$ with capacities given by $c'(e) = c(e) - \sum_{f \in F} f(e)$.*

We will list without proof some well-known facts about maximum flows and minimum cuts in a graph which will be useful in our arguments.

**Fact.** *1. Let $f$ be any feasible $(s, t)$-flow, and let $(S, T)$ be any $(s, t)$-cut. The value of $f$ is at most the capacity of $(S, T)$. Moreover, the value of $f$ equals the capacity of $(S, T)$ if and only if $f$ saturates every edge in the cut.*

*2. Max-flow min-cut theorem. The value of maximum (value of) $(s, t)$-flow equals the capacity of the minimum $(s, t)$-cut. It may be computed in $O(VE)$ time.*

*3. Flow Decomposition Theorem. Every feasible $(s, t)$-flow $f$ can be written as a weighted sum of directed $(s, t)$-paths and directed cycles. Moreover, a directed edge $(u, v)$ appears in at least one of these paths or cycles if and only if $f(u, v) > 0$, and the total number of paths and cycles is at most the number of edges in the network. It may be computed in $O(VE)$ time.*

---

**Algorithm 3** DYNAMICMINCUT$(G, \sigma_0, \epsilon)$

---

1: **Input:** Graph $G$ with unlabeled nodes, query parameter $\sigma_0$, error tolerance $\epsilon$.
2: **Output:** Piecewise constant interval containing $\sigma_0$.
3: Use a max-flow algorithm to compute max-flow and min-cut $\mathcal{C}$ for $G(\sigma)$, $\sigma_h = \sigma_0$.
4: Compute the flow decomposition of the max-flow, $\mathcal{F}$.
5: Let $f_e$ be a unique *path flow* (i.e. along an $st$-path, Definition 27) through $e \in \mathcal{C}$.
6: Say $e$ is *augmentable* if flow $f_e$ can be increased by amount $w_e(\sigma) - w_e(\sigma_h)$ for some $\sigma > \sigma_h$.
   $e$ acts as the bottleneck for increasing the flow $f_e$.
7: Initialize $S$ to $\mathcal{C}$ (a set of saturated edges).
8: **while** All edges $e \in S$ are augmentable, **do**
9:     Increase flow in all $f_e$ for $e \in S$ to keep $e$ saturated.
10:    Find first saturating edge $e_1 \notin S$ for some $f_{e'}$ ($e' \in S$) and $\sigma'$ to within $\epsilon$.
11:    Reassociate flow through $e_1, e'$ as $f_{e_1}$. $f_{e'}$ will now be along an alternate path in the residual
       capacities graph (Definition 28).
12:    Add $e_1$ to $S$.
13:    Set $\sigma_h = \sigma'$.
14: Similarly find the start of the interval $\sigma_l$ by detecting saturation while reducing flows.
15: **return** $[\sigma_l, \sigma_h]$.

---

We now have the machinery to prove the correctness and analyze the time complexity of our Algorithm 3.

**Theorem 9.** *For the each objective in Table 1 and exponential kernel (Definition 1c), there exists an algorithm which outputs the interval containing $\sigma$ in time $\tilde{O}(n^4)$.*

*Proof. Mincut objective.* First, we briefly recall the set up of the mincut objective. Let $L_1$ and $L_2$ denote the labeled points $L$ of different classes. To obtain the labels for $U$, we seek the smallest cut $(V_1, V \setminus V_1)$ of $G$ separating the nodes in $L_1$ and $L_2$. To frame as $s, t$-cut we can augment the data graph with nodes $s, t$, and add infinite capacity edges to nodes in $L_1$ and $L_2$ respectively. If $L_i \subseteq V_1$, label exactly the nodes in $V_1$ with label $i$. The loss function, $l(\sigma)$ gives the fraction of labels this procedure gets right for the unlabeled set $U$. We now discuss the correctness of Algorithm 3.

If the min-cut is the same for two values of $\sigma$, then so is prediction on each point and thus the loss function $l(\sigma)$. So we seek the smallest amount of change in $\sigma$ so that the mincut changes. Our semi-bandit feedback set is given by the intervals for which the min-cut is fixed. Consider a fixed value of $\sigma = \sigma_0$ and the corresponding graph $G(\sigma_0)$. We can compute the max-flow on $G(\sigma_0)$, and simultaneously obtain a min-cut $(V_1, V \setminus V_1)$ in time $O(VE) = O(n^3)$. All the edges in $\partial V_1$ are saturated by the flow. Obtain the flow decomposition of the max-flow (again $O(VE) = O(n^3)$). For each $e_i \in \partial V_1$, let $f_i$ be a path flow through $e_i$ from the flow decomposition (cycle flows cannot saturate, or even pass through, $e_i$ since it is on the min-cut). Note that the $f_i$ are distinct due to the max-flow min-cut theorem. Now as $\sigma$ is increased, we increment each $f_i$ by the additional capacity in the corresponding edge $e_i$, until an edge $e'$ in $E \setminus \partial V_1$ saturates (at a faster rate than the flow through it). This can be detected by expressing $f_i$ as a function of $\sigma$ for each $f_i$ and computing the zero of an exponential polynomial capturing the change in residual capacity of any edge $e \notin \partial V_1$. Let $f_j$ be one of the path flows through $e'$. We reassign this flow to $e'$ (it will now increase with $e'$ as its bottleneck) and find an alternate path avoiding this edge through non-saturated edges and $e_j$ (if one exists) along which we send the new $f_j$. We now increment all the path flows as before keeping their bottleneck edges saturated. The procedure stops when we can no longer find an alternate path for some $e_j$. But this means we must have a new cut with the saturated edges, and therefore a new min-cut. This gives us a new critical value of $\sigma$, and the desired upper end for the feedback interval. Obtaining the lower end is possible by a symmetric procedure that decreases the path flows while keeping edges saturated.

We remark that our procedure differs from the well-known algorithms for obtaining min-cuts in a static graph. The greedy procedures for static graphs need directed edges $(u, v)$ and $(v, u)$ in the residual graph, and find paths through unsaturated edges through this graph to increase the flow, and cannot work with monotonically increasing path flows. We however start with a max flow and maintain the invariant that our flow equals some cut size throughout.

Finally note that each time we perform step 9 of the algorithm, a new saturated edge stays saturated for all further $\sigma$ until the new cut is found. So we can do this at most $O(n^2)$ times. In each loop we need to obtain the saturation condition for $O(n)$ edges corresponding to one new path flow. Thus the entire procedure takes $O(n^3 K(n, \epsilon))$ time, where $K(n, \epsilon)$ is the complexity of solving an exponential equation $\phi(y) = \sum_{i=1}^n a_i y^{b_i} = 0$ to within error $\epsilon$. For example, $K(n, \epsilon)$ is $O(n \log \log \frac{1}{\epsilon})$ for the Newton's method.

*Other objectives.* For continuous objectives, we seek a solution $f_u(\sigma) = \frac{1}{2}$ for some $u \in U$ closest to given $\sigma_0$. We can use gradient descent or Newton's method with $\sigma_0$ as the starting point. For the harmonic objective the gradient may be computed as in Appendix C.5.2. □

We remind the reader that a remarkable property of finding the min-cuts dynamically in our setting is an interesting "hybrid" combinatorial and continuous set-up, which may be of independent interest. A similar dynamic, but purely combinatorial, setting for recomputing flows efficiently online over a discrete graph sequence has been studied in Altner and Ergun [2008].

### C.5.2 Quadratic objective

For completeness we describe how to compute the feedback set for the quadratic objective using Newton's method. Running time guarantees are noted in Theorem 9.

---

**Algorithm 4** HARMONICFEEDBACKSET$(G, \sigma_0, \epsilon)$

---

1: **Input:** Graph $G$ with unlabeled nodes, query parameter $\sigma_0$, error tolerance $\epsilon$.
2: **Output:** Piecewise constant interval containing $\sigma_0$.
3: Let $f_U = (D_{UU} - W_{UU})^{-1} W_{UL} f_L$ denote the harmonic objective minimizer, where $D_{ij} := \mathbb{I}[i = j] \sum_k W_{ik}$.
4: **for all** $u \in U$ **do**
5:     Let $g_u(\sigma) = (f_u(\sigma) - \frac{1}{2})^2$.
6:     $\sigma_1 = \sigma_0 - \frac{g_u(\sigma_0)}{g'_u(\sigma_0)}$, where $g'_u(\sigma)$ is given by

$$\frac{\partial g_u}{\partial \sigma} = 2\left(f_u(\sigma) - \frac{1}{2}\right)\frac{\partial f_u}{\partial \sigma},$$

$$\frac{\partial f}{\partial \sigma} = (I - P_{UU}^{-1})\left(\frac{\partial P_{UU}}{\partial \sigma} f_U - \frac{\partial P_{UL}}{\partial \sigma} f_L\right),$$

$$\frac{\partial P_{ij}}{\partial \sigma} = \frac{\frac{\partial w(i,j)}{\partial \sigma} - P_{ij}\sum_{k \in L+U}\frac{\partial w(i,k)}{\partial \sigma}}{\sum_{k \in L+U} w(i,k)},$$

$$\frac{\partial w(i,j)}{\partial \sigma} = \frac{2w(i,j)d(i,j)^2}{\sigma^3},$$

    where $P = D^{-1}W$.
7:     $n = 0$
8:     **while** $|\sigma_{n+1} - \sigma_n| \geq \epsilon$ **do**
9:         $n \leftarrow n + 1$
10:        $\sigma_{n+1} = \sigma_n - \frac{g_u(\sigma_n)}{g'_u(\sigma_n)}$
11:    $\sigma_u = \sigma_{n+1}$
12: $\sigma_l = \max_u\{\sigma_u \mid \sigma_u < \sigma_0\}$, $\sigma_h = \min_u\{\sigma_u \mid \sigma_u > \sigma_0\}$
13: **return** $[\sigma_l, \sigma_h]$.

---

## D   Distributional setting

In this appendix we include details of proofs and algorithms from section 5. Recall that we define the set of loss functions $\mathcal{H}_r = \{l_{A(G(r), L, U)} \mid 0 \leq r < \infty\}$, where $G(r)$ is the family of threshold graphs specified by Definition 1a, and $\mathcal{H}_\sigma = \{l_{A(G(\sigma), L, U)} \mid 0 \leq \sigma < \infty\}$, where $G(\sigma)$ is the family of exponential kernel graphs specified by Definition 1c. We show upper and lower bounds on the pseudodimension of these function classes below.

**Theorem 10.** *The pseudo-dimension of $\mathcal{H}_r$ is $\Theta(\log n)$, where $n$ is number of graph nodes.*

*Proof.* There are at most $\binom{n}{2}$ distinct distances between pairs of data points. As $r$ is increased from 0 to infinity, the graph changes only when $r$ corresponds to one of these distances, and so at most $\binom{n}{2} + 1$ distinct graphs may be obtained.

Thus given set $\mathcal{S}$ of $m$ instances $(A^{(i)}, L^{(i)})$, we can partition the real line into $O(mn^2)$ intervals such that all values of $r$ behave identically for all instances within any fixed interval. Since $A$ and therefore its loss is deterministic once $G$ is fixed, the loss function is a piecewise constant with only $O(n^2)$ pieces. Each piece can have a witness above or below it as $r$ is varied for the corresponding interval, and so the binary labeling of $\mathcal{S}$ is fixed in that interval. The pseudo-dimension $m$ satisfies $2^m \leq O(mn^2)$ and is therefore $O(\log n)$.

To establish the lower bound, we first prove the following useful statement which helps us construct general examples with desirable properties. In particular, the following lemma guarantees that given a sequence of values of $r$ of size $O(n)$, it is possible to construct an instance $S$ of partially labeled points such that the cost of the output of algorithm $A(G(r), L)$ on V as a function of $r$ oscillates above and below some threshold as $r$ moves along the sequence of intervals $(r_i, r_{i+1})$. Given this powerful guarantee, we can then pick appropriate sequences of $r$ and generate a sample set of $\Omega(\log n)$ instances that correspond to cost functions that oscillate in a manner that helps us pick $\Omega(n)$ values of $r$ that shatters the samples.

**Lemma 29.** *Given integer $n > 5$ and a sequence of $n'$ $r$'s such that $1 < r_1 < r_2 < \cdots < r_{n'} < 2$ and $n' \leq n-5$, there exists a real valued witness $w > 0$ and a labeling instance $S$ of partially labeled $n$ points, such that for $0 \leq i \leq n'/2 - 1$, $l_{A(G(r),L)} < w$ for $r \in (r_{2i}, r_{2i+1})$, and $l_{A(G(r),L)} > w$ for $r \in (r_{2i+1}, r_{2i+2})$ (where $r_0$ and $r_{n'+1}$ correspond to immediate left and right neighborhoods respectively of $r_1$ and $r_{n'}$).*

*Proof.* We first present a sketch of the construction. We will use binary labels $a$ and $b$. We further have three points labeled $a$ (namely $a_1, a_2, a_3$) and two points labeled $b$ (say $b_1, b_2$). At some intial $r = r_0$, all the like-labeled points are connected in $G(r_0)$ and all the unlabeled points (namely $u_1, \ldots, u_{n'}$) are connected to $a_1$ as shown in Figure 4a. The algorithm $A(G(r), L)$ labels everything $a$ and gets exactly half the labels right. As $r$ is increased to $r_i$, $u_i$ gets connected to $b_1$ and $b_2$ (Figure 4b). If the sequence $u_i$ is alternately labeled, the loss increases and decreases alternately as all the predicted labels turn to $b$ as $r$ is increased to $r_{n'}$. Further increasing $r$ may connect all the unlabeled points with true label $a$ to $a_2$ and $a_3$ (Figure 4c), although this is not crucial to our argument. The rest of the proof gives concrete values of $r$ and verifies that the construction is indeed feasible.

We will ensure all the pairwise distances are between 1 and 2, so that triangle inequality is always satisfied. It may also be readily verified that $O(\log n)$ dimensions suffice for our construction to exist. We start by defining some useful constants. We pick $r_-, r_+, r_{\max} \in (1, 2)$ such that $r_- < r_1 < \cdots < r_{n'} < r_+ < r_{\max}$,

$$r_- = \frac{1 + r_1}{2},$$
$$r_+ = 1 + \frac{r_{n'}}{2},$$
$$r_{\max} = 1 + \frac{r_+}{2}.$$

We will now specify the distances of the labeled points. The points with the same label are close together and away from the oppositely labeled points.

$$
\begin{aligned}
d(a_i, a_j) &= r_-, & 1 \leq i < j \leq 3, \\
d(b_1, b_2) &= r_-, & \\
d(a_i, b_j) &= r_{\max}, & 1 \leq i \leq 3, 1 \leq j \leq 2.
\end{aligned}
$$

Further, the unlabeled points are located as follows

$$
\begin{aligned}
d(a_1, u_k) &= r_-, & 1 \leq k \leq n', \\
d(b_i, u_k) &= r_k, & 1 \leq k \leq n', 1 \leq i \leq 2, \\
d(a_i, u_k) &= r_+, & 1 \leq k \leq n', 2 \leq i \leq 3, \\
d(u_i, u_j) &= r_{\max}, & 1 \leq i < j \leq n'.
\end{aligned}
$$

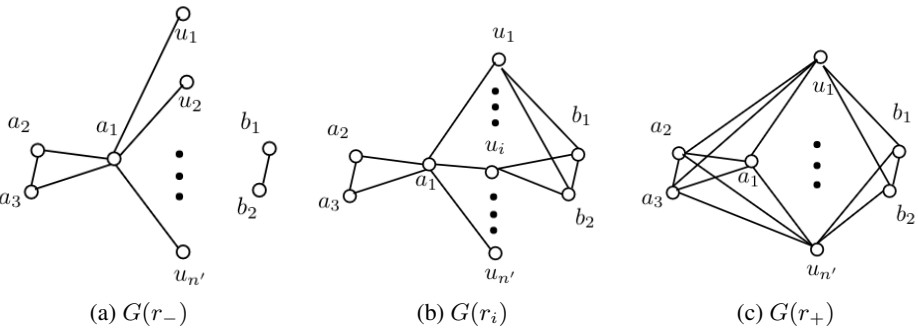

$$\text{(a) } G(r_-) \qquad\qquad \text{(b) } G(r_i) \qquad\qquad \text{(c) } G(r_+)$$

Figure 4: Graphs $G(r)$ as $r$ is varied, for lower bound construction for pseudodimension of $\mathcal{H}_r$.

That is, all unknown points are closest to $a_1$, followed by $b_i$'s, remaining $a_i$'s and other $u_i$'s in order. Further let the true labels of the unlabeled nodes be alternating with the index, i.e. $u_k$ is $a$ if and only if $k$ is even.

We will now compute the loss for the soft labeling algorithm $A(G(r), L)$ of Zhu et al. [2003] as $r$ varies from $r_-$ to $r_+$, starting with $r = r_0 = r_-$. We note that our construction also works for other algorithms as well, for example the min-cut based approach of Blum and Chawla [2001], but omit the details.

For the graph $G(r_-)$, $A(G, L)$ labels each unknown node as $a$ since each unknown point is a leaf node connected to $a_1$. Indeed if $f(a_1) = 1$, the quadratic objective attains the minimum of 0 for exactly $f(u_k) = 1$ for each $1 \le k \le n'$. This results in half the labels in the dataset being incorrectly labeled since we stipulate that half the unknown labels are of each category. This results in loss $l_{A(G(r_-),L)} =: l_{\text{high}}$ say.

Now as $r$ is increased to $r_1$, the edges $(b_i, u_1)$, $i = 1, 2$ are added with $b_i$ labeled as $f(b_i) = 0$. This results in a fractional label of $\frac{1}{3}$ for $f(u_1)$ while $f(u_k) = 1$ for $k \ne 1$. Indeed the terms involving $f(u_1)$ in the objective are $(1 - f(u_1))^2 + 2f(u_1)^2$, which is minimized at $\frac{1}{3}$. Since $u_1$ has true label $b$, this results in a slightly smaller loss of $l_{A(G(r_1),L)} =: l_{\text{low}}$. This happens when $A$ uses rounding, or in expectation if $A$ uses randomized prediction with probability $f(u)$.

At the next critical point $r_2$, $u_2$ gets connected to $b_i$'s and gets incorrectly classified as $b$. This increases the loss again to $l_{\text{high}}$. The loss function thus alternates as $r$ is varied through the specified values, between $l_{\text{high}}$ and $l_{\text{low}}$. We therefore set the witness $w$ to something in between.

$$w = \frac{l_{\text{low}} + l_{\text{high}}}{2}.$$

$\square$

*Continued Proof of Theorem 10* We will now use Lemma 29 to prove our lower bound. Arbitrarily choose $n' = n - 5$ (assumed to be a power of 2 just for convenient presentation) real numbers $r_{[000...01]} < r_{[000...10]} < \cdots < r_{[111...11]}$ in $(1, 2)$. The indices are increasing binary numbers of length $m = \log n'$. We create labeling instances using Lemma 29 which can be shattered by these $r$ values. Instance $S_i = (G_i, L_i)$ corresponds to fluctuation of $i$-th bit $b_i$ in our $r_b$ sequence, where $b = (b_1, \ldots, b_m) \in \{0, 1\}^m$, i.e., we apply the lemma by using a subset of the $r_b$ values which correspond to the bit flips in the $i$-th binary digit. For example, $S_1$ just needs a single bit flip (at $r_{[100...00]}$). The lemma gives us both the instances and corresponding witnesses $w_i$.

This construction ensures $\text{sign}(l_{A(G_i(r_b),L_i)} - w_i) = b_i$, i.e. the set of instances is shattered. Thus the pseudodimension is at least $\log(n - 5) = \Omega(\log n)$. $\square$

**Theorem 11.** *The pseudo-dimension of $\mathcal{H}_\sigma$ is $\Theta(n)$.*

*Proof.* The upper bound trivially follows by noting that we have $n$ vertices and therefore only $2^n$ possible labelings in an instance. Thus, for $m$ problems, $2^m \le m2^n$ gives $m = O(n)$. The rest of the proof deals with the lower bound.

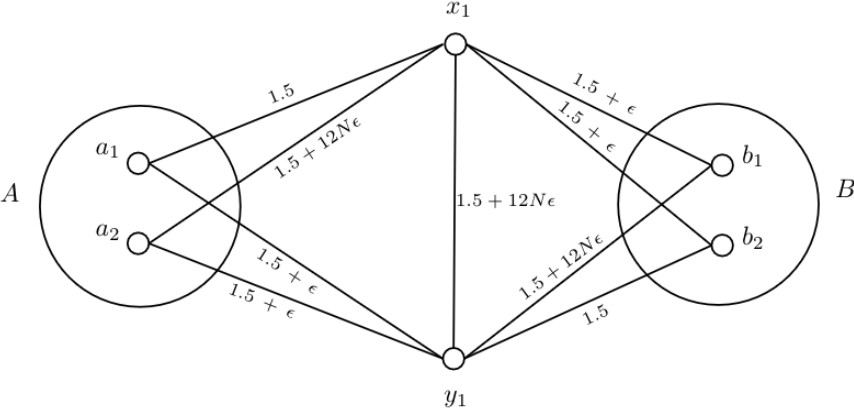

Figure 5: The base case of our inductive construction.

The plan for the proof is to first construct a graph where the edge weights are carefully selected, so that we have $2^N$ oscillations in the loss function with $\sigma$ for $N = \Omega(n)$. Then we use this construction to create $\Theta(n)$ instances, each having a subset of the oscillations so that each interval leads to a unique labeling of the instances, for a total of $2^N$ labelings, which would imply pseudodimension is $\Omega(n)$. We will present our discussion in terms of the min-cut objective, for simplicity of presentation.

*Graph construction*: First a quick rough overview. We start with a pair of labeled nodes of each class, and a pair of unlabeled nodes which may be assigned either label depending on $\sigma$. We then build the graph in $N = (n-4)/2$ stages, adding two new nodes at each step with carefully chosen distances from existing nodes. Before adding the $i$th pair $x_i, y_i$ of nodes, there will be $2^{i-1}$ intervals of $\sigma$ such that the intervals correspond to distinct min-cuts which result in all possible labelings of $\{x_1, \ldots, x_{i-1}\}$. Moreover, $y_j$ will be labeled differently from $x_j$ in each of these intervals. The edges to the new nodes will ensure that the cuts that differ exactly in $x_i$ will divide each of these intervals giving us $2^i$ intervals where distinct mincuts give all labelings of $\{x_1, \ldots, x_i\}$, and allowing an inductive proof. The challenge is that we only get to set $O(i)$ edges but seek properties about $2^i$ cuts, so we must do this carefully.

Let $\varsigma = e^{-1/\sigma^2}$. Notice $\varsigma \in (0, 1)$, and bijectively corresponds to $\sigma \in (0, \infty)$ (due to monotonicity) and therefore it suffices to specify intervals of $\varsigma$ corresponding to different labelings. Further we can specify distances $d(u, v)$ between pairs of nodes $u, v$ by specifying the squared distance $d(u, v)^2$. For the remainder of this proof we will refer to $\delta(u, v) = d(u, v)^2$ by *distance* and set values in $[1.5, 1.6]$. Consequently, $d(u, v) \in (1.22, 1.27)$ and therefore the triangle inequality is always satisfied. Notice that with this notation, the graph weights will be $w(u, v) = \varsigma^{\delta(u, v)}$.

We now provide details of the construction. We have four labeled nodes as follows. $a_1, a_2$ are labeled $0$ and are collectively denoted by $A = \{a_1, a_2\}$, similarly $b_1, b_2$ are labeled $1$ and $B = \{b_1, b_2\}$. Note that edges between these nodes are on all or no cut separating $A, B$, we set the distances to $1.6$ and call this graph $G_0$. We further add unlabeled nodes in pairs $(x_j, y_j)$ in *rounds* $1 \le j \le N$. In round $i$, we construct graph $G_i$ by adding nodes $(x_i, y_i)$ to $G_{i-1}$. The distances are set to ensure that for $G_N$ there are $2^N$ unique values of $\varsigma$ corresponding to distinct min-cuts, each giving a unique labeling for $\{x_1, \ldots, x_n\}$ (and the complementary labeling for $\{y_1, \ldots, y_n\}$). Moreover subsets of these points also obtain the unique labeling for $\{x_1, \ldots, x_i\}$ for each $G_i$.

We set the distances in round 1 such that there are intervals $I_0 = (\varsigma_0, \varsigma_0') \subset (0, 1)$ and $I_1 = (\varsigma_1, \varsigma_1') \subset (0, 1)$ such that $\varsigma_0' < \varsigma_1$ and $(x_1, y_1)$ are labeled $(l, 1-l)$ in interval $I_l$. In general, there will be $2^{i-1}$ intervals at the end of round $i-1$, any interval $I^{(i-1)}$ will be split into disjoint intervals $I_0^{(i)}, I_1^{(i)} \subset I^{(i-1)}$ where labelings of $\{x_1, \ldots, x_{i-1}\}$ match that of $I^{(i-1)}$ and $(x_i, y_i)$ are labeled $(l, 1-l)$ in $I_l^{(i)}$.

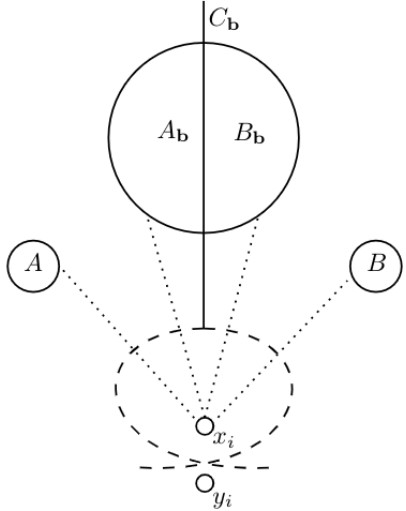

Figure 6: The inductive step in our lower bound construction for pseudodimension of $\mathcal{H}_\sigma$. The min-cut $C_{\mathbf{b}}$ is extended to two new min-cuts (depicted by dashed lines) for which labels of $x_i, y_i$ are flipped, at controlled parameter intervals.

Now we set up the edges to achieve these properties. In round 1, we set the distances as follows.

$$\delta(x_1, a_1) = \delta(y_1, b_2) = 1.5,$$
$$\delta(x_1, a_2) = \delta(y_1, b_1) = \delta(x_1, y_1) = 1.5 + 12N\epsilon,$$
$$\delta(x_1, b_1) = \delta(x_1, b_2) = \delta(y_1, a_1) = \delta(y_1, a_2) = 1.5 + \epsilon.$$

where $\epsilon$ is a small positive quantity such that the largest distance $1.5 + 12N\epsilon < 1.6$. It is straightforward to verify that for $I_0 = (0, \frac{1}{2}^{1/\epsilon})$ we have that $(x_1, y_1)$ are labeled $(0, 1)$ by determining the values of $\varsigma$ for which the corresponding cut is the min-cut (Figure 5). Indeed, we seek $\varsigma$ such that $w_{C01} = w(x_1, b_1) + w(x_1, b_2) + w(x_1, y_1) + w(y_1, a_1) + w(y_1, a_2)$ satisfies

$$w_{C01} \leq w_{C00} = w(x_1, b_1) + w(x_1, b_2) + w(y_1, b_1) + w(y_1, b_2),$$

$$w_{C01} \leq w_{C11} = w(x_1, a_1) + w(x_1, a_2) + w(y_1, a_1) + w(y_1, a_2),$$

$$w_{C01} \leq w_{C10} = w(x_1, a_1) + w(x_1, a_2) + w(x_1, y_1) + w(y_1, b_1) + w(y_1, b_2),$$

which simultaneously hold for $\varsigma < \frac{1}{2}^{1/\epsilon}$.

Moreover, we can similarly conclude that $(x_1, y_1)$ are labeled $(1, 0)$ for the interval $I_1 = (\varsigma_1, \varsigma_1')$ where $\varsigma_1 < \varsigma_1'$ are given by the two positive roots of the equation

$$1 - 2\varsigma^\epsilon + 2\varsigma^{12N\epsilon} = 0.$$

We now consider the inductive step, to set the distances and obtain an inductive proof of the claim above. In round $i$, the distances are as specified.

$$\delta(x_i, a_1) = \delta(y_i, b_2) = 1.5,$$
$$\delta(x_i, a_2) = \delta(y_i, b_1) = \delta(x_i, y_i) = 1.5 + 12N\epsilon,$$
$$\delta(x_i, b_1) = \delta(x_i, b_2) = \delta(y_i, a_1) = \delta(y_i, a_2) = 1.5 + \epsilon,$$
$$\delta(x_i, y_j) = \delta(y_i, x_j) = 1.5 + 6(2j - 1)\epsilon \quad (1 \leq j \leq i - 1),$$
$$\delta(x_i, x_j) = \delta(y_i, y_j) = 1.5 + 12j\epsilon \quad (1 \leq j \leq i - 1).$$

We denote the (inductively hypothesized) $2^{i-1}$ $\varsigma$-intervals at the end of round $i - 1$ by $I_{\mathbf{b}}^{(i-1)}$, where $\mathbf{b} = \{b^{(1)}, \ldots, b^{(i-1)}\} \in \{0, 1\}^{i-1}$ indicates the labels of $x_j, j \in [i - 1]$ in $I_{\mathbf{b}}^{(i-1)}$. Min-cuts from round $i - 1$ extend to min-cuts of round $i$ depending on how the edges incident on $(x_i, y_i)$ are set (Figure 6). It suffices to consider only those min-cuts where $x_j$ and $y_j$ have opposite labels for each

$j$. Consider an arbitrary such min-cut $C_{\mathbf{b}} = (A_{\mathbf{b}}, B_{\mathbf{b}})$ of $G_{i-1}$ which corresponds to the interval $I_{\mathbf{b}}^{(i-1)}$, that is $A_{\mathbf{b}} = \{x_j \mid b^{(j)} = 0\} \cup \{y_j \mid b^{(j)} = 1\}$ and $B_{\mathbf{b}}$ contains the remaining unlabeled nodes of $G_{i-1}$. It extends to $C_{[\mathbf{b}\ 0]}$ and $C_{[\mathbf{b}\ 1]}$ for $\varsigma \in I_{\mathbf{b}}^{(i-1)}$ satisfying, respectively,

$$E_{\mathbf{b},0}(\varsigma) := 1 - 2\varsigma^\epsilon + F(C_{\mathbf{b}};\varsigma) > 0,$$
$$E_{\mathbf{b},1}(\varsigma) := 1 - 2\varsigma^\epsilon + 2\varsigma^{12N\epsilon} + F(C_{\mathbf{b}};\varsigma) < 0,$$

where $F(C_{\mathbf{b}};\varsigma) = \sum_{z \in A_{\mathbf{b}}} \varsigma^{\delta(x_i,z)} - \sum_{z \in B_{\mathbf{b}}} \varsigma^{\delta(x_i,z)} = \sum_{z \in B_{\mathbf{b}}} \varsigma^{\delta(y_i,z)} - \sum_{z \in A_{\mathbf{b}}} \varsigma^{\delta(y_i,z)}$. If we show that the solutions of the above inequations have disjoint non-empty intersections with $\varsigma \in I_{\mathbf{b}}^{(i-1)}$, our induction step is complete. We will use an indirect approach for this.

For $1 \leq i \leq N$, given $\mathbf{b} = \{b^{(1)},\ldots,b^{(i-1)}\} \in \{0,1\}^{i-1}$, let $E_{\mathbf{b},0}$ and $E_{\mathbf{b},1}$ denote the expressions (exponential polynomials in $\varsigma$) in round $i$ which determine labels of $(x_i, y_i)$, in the case where for all $1 \leq j < i$, $x_j$ is labeled $b^{(j)}$ (and let $E_{\phi,0}, E_{\phi,1}$ denote the expressions for round 1). Let $\varsigma_{\mathbf{b},i} \in (0,1)$ denote the smallest solution to $E_{\mathbf{b},i} = 0$. Then we need to show the $\varsigma_{\mathbf{b},i}$'s are well-defined and follow a specific ordering. This ordering is completely specified by two conditions:

   (i) $\varsigma_{[\mathbf{b}\ 0],1} < \varsigma_{[\mathbf{b}],0} < \varsigma_{[\mathbf{b}],1} < \varsigma_{[\mathbf{b}\ 1],0}$, and

   (ii) $\varsigma_{[\mathbf{b}\ 0\ \mathbf{c}],1} < \varsigma_{[\mathbf{b}\ 1\ \mathbf{d}],0}$

for all $\mathbf{b}, \mathbf{c}, \mathbf{d} \in \cup_{i<N}\{0,1\}^i$ and $|\mathbf{c}| = |\mathbf{d}|$.

First we make a quick observation that all $\varsigma_{\mathbf{b},i}$'s are well-defined and less than $(3/4)^{1/\epsilon}$. To do this, it will suffice to note that $E_{\mathbf{b},i}(0) = 1$ and $E_{\mathbf{b},i}(\frac{3}{4}^{1/\epsilon}) < 0$ for all $\mathbf{b}, i$, since the functions are continuous in $(0, \frac{3}{4}^{1/\epsilon})$. This holds because

$$E_{\mathbf{b},0}\left(\frac{3}{4}^{1/\epsilon}\right) < E_{\mathbf{b},1}\left(\frac{3}{4}^{1/\epsilon}\right) = 1 - \frac{3}{2} + \left(\frac{3}{4}\right)^{12N} + F\left(C_{\mathbf{b}}; \frac{3}{4}^{1/\epsilon}\right)$$
$$\leq -\frac{1}{2} + \left(\frac{3}{4}\right)^{12N} + \sum_{j=1}^{|\mathbf{b}|}\left(\frac{3}{4}\right)^{6j}\left(1 - \left(\frac{3}{4}\right)^{6j}\right)$$
$$< -\frac{1}{2} + \sum_{j=1}^{N}\left(\frac{3}{4}\right)^{6j}$$
$$< 0$$

Let's now consider condition (i). We begin by showing $\varsigma_{[\mathbf{b}],0} < \varsigma_{[\mathbf{b}],1}$ for any $\mathbf{b}$. The exponential polynomials $E_{\mathbf{b},0}$ and $E_{\mathbf{b},1}$ both evaluate to 1 for $\varsigma = 0$ (since $|A_{\mathbf{b}}| = |B_{\mathbf{b}}| = |\mathbf{b}|$) and decrease monotonically (verified by elementary calculus) till their respective smallest zeros $\varsigma_{[\mathbf{b}],0}, \varsigma_{[\mathbf{b}],1}$. But then $E_{\mathbf{b},1}(\varsigma_{[\mathbf{b}],0}) = 2(\varsigma_{[\mathbf{b}],0})^{12N\epsilon} > 0$, which implies $\varsigma_{[\mathbf{b}],0} < \varsigma_{[\mathbf{b}],1}$. Now, to show $\varsigma_{[\mathbf{b}\ 0],1} < \varsigma_{[\mathbf{b}],0}$, note that $E_{[\mathbf{b}\ 0],1}(\varsigma) - E_{[\mathbf{b}],0}(\varsigma) = 2\varsigma^{12N\epsilon} + \varsigma^{12i\epsilon} - \varsigma^{(12i-6)\epsilon} = \varsigma^{(12i-6)\epsilon}(2\varsigma^{(12(N-i)+6)\epsilon} + \varsigma^{6\epsilon} - 1)$ where $1 \leq i = |\mathbf{b}| + 1 < N$. Since $\varsigma_{[\mathbf{b}],0} < \frac{3}{4}^{1/\epsilon}$, it follows that $E_{[\mathbf{b}\ 0],1}(\varsigma_{[\mathbf{b}],0}) < 0$, which implies $\varsigma_{[\mathbf{b}\ 0],1} < \varsigma_{[\mathbf{b}],0}$. Similarly, it is readily verified that $\varsigma_{[\mathbf{b}],1} < \varsigma_{[\mathbf{b}\ 1],0}$, establishing (i).

Finally, to show (ii), note that $E_{[\mathbf{b}\ 0\ \mathbf{c}],1}(\varsigma) - E_{[\mathbf{b}\ 0\ \mathbf{d}],0}(\varsigma) = 2\varsigma^{12N\epsilon} + \varsigma^{12i\epsilon} - \varsigma^{(12i-6)\epsilon} + \varsigma^{12i\epsilon}(F(C_{\mathbf{c}};\varsigma) - F(C_{\mathbf{d}};\varsigma)) = \varsigma^{(12i-6)\epsilon}(2\varsigma^{(12(N-i)+6)\epsilon} + \varsigma^{6\epsilon} - 1 + \varsigma^{6\epsilon}(F(C_{\mathbf{c}};\varsigma) - F(C_{\mathbf{d}};\varsigma)))$. Again, similar to above, we use $\varsigma_{[\mathbf{b}\ 0\ \mathbf{d}],0} < \frac{3}{4}^{1/\epsilon}$ in this expression to get $E_{[\mathbf{b}\ 0\ \mathbf{c}],1}(\varsigma_{[\mathbf{b}\ 0\ \mathbf{d}],0}) < 0$. Since the exponential polynomials decay monotonically with $\varsigma$ till their first roots, (ii) follows.

*Problem instances*: We will now show the graph instances and witnesses to establish the pseudodimension bound. Our graphs will be $G_i$ from the above construction (padded appropriately such that the min-cut intervals do not change, if we insist each instance has exactly $n$ nodes), and the shattering family $\sigma_b$ ($b = (b_1, \ldots, b_N) \in \{0,1\}^N$) will be $2^N$ values of $\sigma$ corresponding to the $2^N$ intervals of $\varsigma$ with distinct min-cuts in $G_N$ described above. To obtain the witnesses, we set the labels so that only the last pair of nodes $(x_i, y_i)$ have different labels (i.e. labels are same for all $(x_j, y_j), j < i$)

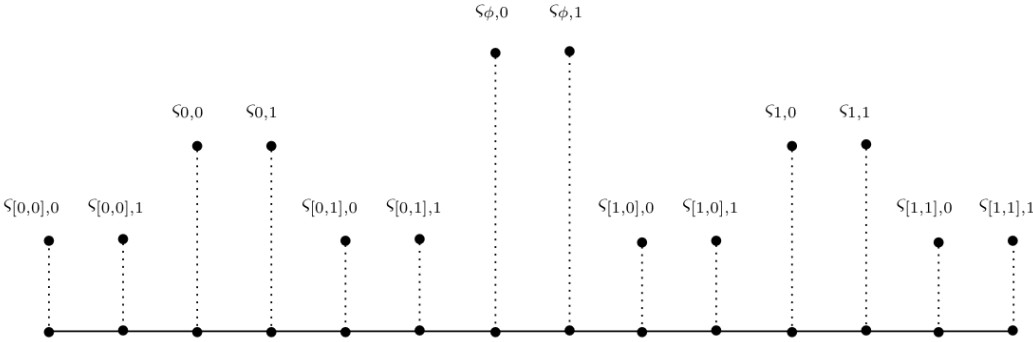

Figure 7: Relative positions of critical values of the parameter $\varsigma = e^{-1/\sigma^2}$.

and therefore the loss function oscillates $2^i$ times as $(x_i, y_i)$ are correctly and incorrectly labeled in alternating intervals. The intervals of successive $G_i$ are nested precisely so that $\sigma_b$ shatter the instances for the above labelings/witnesses. Thus, we have shown that the pseudodimension is $\Omega(N) = \Omega((n-4)/2) = \Omega(n)$. $\qquad\square$

# E    Further experiments

We include plots for variation of loss function with graph hyperparameters $r, \sigma$ for unweighted graphs $G(r)$ and weighted graphs $G(\sigma)$ for single instances of datasets drawn as described in Section 6. We examine the full variation of performance of graph-based semi-supervised learning for all possible graphs $G(r)$ ($r_{\min} < r < r_{\max}$) and $G(\sigma)$ for $\sigma \in [0, 10]$ (Figures 8, 9).

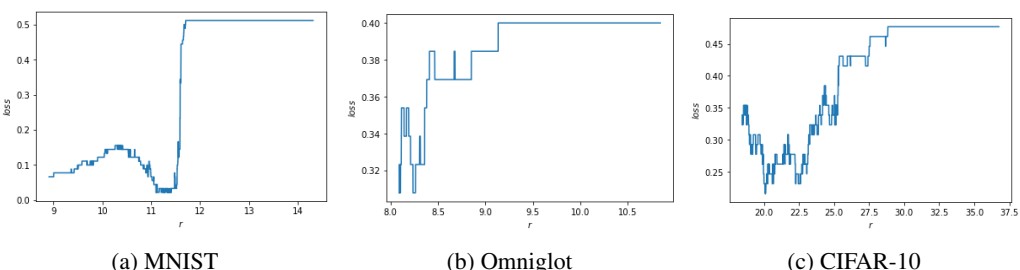

(a) MNIST            (b) Omniglot            (c) CIFAR-10

Figure 8: Loss for different unweighted graphs as a function of the threshold $r$.

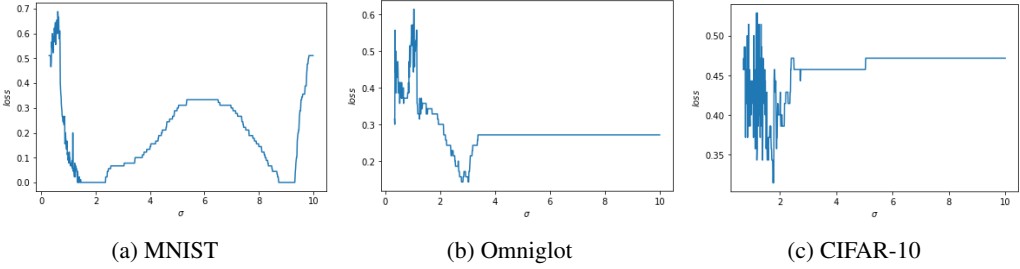

(a) MNIST            (b) Omniglot            (c) CIFAR-10

Figure 9: Loss for different weighted graphs as a function of the parameter $\sigma$.