# OpenReview forum: "Data driven semi-supervised learning"
_NeurIPS.cc/2021/Conference — NeurIPS 2021 Oral_

### Official Review · Reviewer_vHeJ · 2021-07-12

**Rating:** 8
**Confidence:** 2

**Summary:**

The authors propose to learn the underlying graphs for graph-based semi-supervised learning problems. So far, graph-based SSL usually grounds on KNN-like graphs where distances are computed according to some measure. The present paper now learns the parameters of the measures (kernels) as well as a threshold to determine whether an edge is present or not.

**Limitations And Societal Impact:**

Yes in terms of limitations, there are no societal aspects.

**Main Review:**

This is a very interesting paper that introduces solid theoretical results that may have an impact in domains different from SSL. The derivations heavily build on different results by Balcan and colleagues and render the technical contribution very strong and convincing, including novel regret bound, algorithms, as well as generalization guarantees in probabilistic scenarios. The appendix is very helpful and gives many additional insights.

I appreciate the authors response.

**Time Spent Reviewing:**

3

---

> ### Author Response · Authors · 2021-08-11
> **Response to Reviewer vHeJ**
>
> We thank the reviewer for their time reviewing our paper. We appreciate that the reviewer finds our results interesting and significant.

---

### Official Review · Reviewer_YqaQ · 2021-07-16

**Rating:** 6
**Confidence:** 3

**Summary:**

This paper proposes a data driven approach to construct graph for graph-based semi-supervised classification task. The authors realize the graph quality is important to learning performance and they aim to update the hyper-parameter when new samples come. They present the algorithms for two cases, online and distributional settings.

**Limitations And Societal Impact:**

The authors have pointed out one limitation of this work (line 460).

**Main Review:**

Originality: The task of updating hyper-parameter for creating graph is new. The method seems an extension of Balcan's work, but shows some novelty as focusing on different tasks. The related works are properly cited and major contributions are nicely stated.

Quality: I have some concerns by reading this paper.
1. If the number of data (L+U) is already large at time of t, the necessity of updating the hyper-parameter for the same domain instances are reduced? So I would like to know if the same domain indicates that data are drew from the identical distribution. If so, it strictly not online learning.
2. It is observed that a target labeling tau is used for computing the loss l_A (line 111).  What is target labeling? Is it side-information compared to the regularized loss (line 43)? Because I have found tau is used in algorithm (step 7 in Alg 1)
3. For the online setting, it seems every time a new sample comes the whole graph will be updated. Is it practical in real case? How about updating a small fraction of entries of w?
4. I am new to regret loss. Is regret estimated from hindsight?
5. Can we say regret in Fig 3(b) converges to the optimal parameter within 50 iterations?
6. Some equations and symbols. Check inequality line 131; rho \in P (line 108) but rho \in C (line 131); definition of \bar{U} (line 191), etc.
7. I think it is better to say "graph-based SSL" in title.

Clarity: The submission clearly written except some typos.

Significance: As graph based semi-supervised learning is not popular for today's research due to its intrinsic limitations, the extension of the main theory from Balcan et al. for this task is not very interesting for me. But this paper is still valuable for some readers in the community.

**Time Spent Reviewing:**

5h

---

> ### Author Response · Authors · 2021-08-11
> **Response to Reviewer YqaQ**
>
> Thank you for the review and useful feedback. Thank you also for recognizing the novelty of the problem. We are glad to hear that you found the related works are properly cited and major contributions are nicely stated.
>
> We provide several clarifications addressing the reviewers main concerns below.
>
> 1a.\ [*If the number of data (L+U) is already large at time of t, the necessity of updating the hyper-parameter for the same domain instances are reduced?*]
>
> Note that the distribution considered in our work is over generation of problem instances as the 'data' and the 'hypothesis’ learned is an algorithm (instead of just learning a classifier over the label distribution).
>
>  -- In particular we do not assume that the labels across instances follow the same distribution. The problem instances are related in the informal sense of `same domain’ which is further elaborated below, but it does not imply or assume same label distribution.
>
> -- To understand the subtle difference, and to answer your question in another way, note that increasing the data size (L+U) can improve the labeling accuracy of all the graph parameters, but the question of which parameter is optimal (i.e. how to construct the graph) still remains.
>
> 1b.\ [*So I would like to know if the same domain indicates that data are drew from the identical distribution. If so, it strictly not online learning.*]
>
> We have results for both distributional (section 5) and online (section 4) settings. The former models problem instances coming from an iid distribution, there is no such assumption for the latter and the guarantees hold even for adversarial sequences of problem instances. More concretely, note that
>
>  -- Online setting (Section 4): problem instance sequence is worst case, but `same domain` just means that some domain-specific parameter values are likely to outperform a randomly chosen value.
>
>  -- Distributional setting (Section 5): The same domain just means that instances (and therefore corresponding optimal parameters) are correlated (in the sense they come from a distribution over problems, so optimal parameter over the problem distribution is well-defined).
>
> 2.\ [*What is target labeling? Is it side-information compared to the regularized loss (line 43)?*]
>
> Target labeling is the ground truth (correct) labels for the examples (graph nodes). In the full information online setting (lines 197-199), (the true loss corresponding to) target labeling for a semi-supervised learning problem $(G_i, L_i, U_i)$ is revealed after we make our semi-supervised predictions. This is comparable to online learning in the standard literature where the loss function is revealed after prediction in each round (cf. “Online Learning and Online Convex Optimization” Shalev-Shwartz 2011).
>
> 3.\ [*For the online setting, it seems every time a new sample comes the whole graph will be updated. Is it practical in real case? How about updating a small fraction of entries of w?*]
>
> New sample will have new unseen examples. So updating the graph to include new examples will keep increasing the size of the graph and make it computationally harder to optimize. Instead, we build a new (smaller) graph and just update the internal ‘weights’ of algorithm 1 on the different parameter values. As noted in 1a.\ above, we are trying to learn how the graph should be built (by looking at multiple SSL problems from the same domain) as opposed to learning how to classify (where adding a single labeled example would make sense).
>
> 4.\ [*I am new to regret loss. Is regret estimated from hindsight?*]
>
> Yes, regret is estimated from hindsight. That is, regret is with respect to what would be the optimal parameter for building the graph after we have seen all the instances and their labels (defined in line 199). Introductory notes on regret optimization can be found in “Online Learning and Online Convex Optimization”, Shalev-Shwartz 2011.
>
> 5.\ [*Can we say regret in Fig 3(b) converges to the optimal parameter within 50 iterations?*]
>
> In Figure 3(b) the average regret is very small relative to the optimal parameter after 50 iterations. In our theoretical results, there is a quadratic upper bound ($1/\sqrt{T}$) on the convergence rate of the average regret.
>
> The suggestion for highlighting the graph-based aspect better in the abstract and title sounds very useful. We will also fix all the typos pointed out by the reviewer.

---

> > ### Comment · Reviewer_YqaQ · 2021-08-25
> > **Review after reading authors' response**
> >
> > Thanks for answering my questions and the responses are convincing. I do not have further concerns and agree to accept this paper.

---

### Official Review · Reviewer_Evc7 · 2021-07-16

**Rating:** 4
**Confidence:** 4

**Summary:**

The authors suggest using repeated problem instances to learn input graphs for graph-based semi-supervised algorithms. They extend the concept of dispersion of one-dimensional algebraic polynomials to an arbitrary number of dimensions in order to provide uniform convergence guarantees for the gaussian graph kernel in unweighted offline training and a generalization bound for weighted graphs. They also offer a threshold graph approach for online training under the dispersity assumption that includes assurances in the form of a constraint on the algorithm's regret across repetitions. Finally, they show that the suggested offline technique leads to minimal regret across a large number of repetitions T.

**Limitations And Societal Impact:**

Overall remarks:
*Theorem3 is not self-content.
* There are no comments on Theorem4's inequality: What does this outcome (inequality) imply?
* The argument over how to test the dispersion assumption in the sequence of high-dimensional loss functions of graph parameters in practice is unclear.
* The dataset size, n, is maintained relatively modest (n=100) compared to the size of the benchmark datasets that were utilized, most likely due to the high time complexity of the optimization process for learning the graph parameters (MNIST, CIFAR, Omnigot).
* The low generalization of the optimal parameter values determined via transductive training, as shown in the experimental part, might be explained by the limited sample size used in training.


**Main Review:**

The paper is well written however there are a number of issues making that the paper is not ready for publication:

**Time Spent Reviewing:**

4

---

> ### Author Response · Authors · 2021-08-11
> **Response to Reviewer Evc7**
>
> We thank the reviewer for their time spent reviewing our paper. We believe that we solve a very important and difficult  open problem --- providing theoretically principled approaches for building the graphs for semi-supervised learning, which is crucial for future practical success of semi-supervised learning. We hope that through the rebuttal and discussion, we will  convince the reviewer to raise their evaluation --- we believe they have underestimated the technical novelty and difficulty of our results.
>
>  We aim  to provide additional clarifications below.
>
> Q1. “Theorem3 is not self-content.”
>
> A1. The statement of theorem 3 is self-contained. It assumes familiarity with definitions of shattering (well-known in learning theory) and algebraic hypersurfaces, which are provided for completeness (along with the full proof) in Appendix C.4 in the supplement material.
>
> Q2. “There are no comments on Theorem4's inequality: What does this outcome (inequality) imply?”
>
> A2. Theorem 4: The implication is that it provides a sufficient condition for establishing dispersion for $p$-dimensional functions (with algebraic discontinuities) as noted in lines 160-162. Further context can be obtained by reading our comments about prior work (lines 133-139). More technical details for how to employ the result can be found in the proof of Theorem 8 (the result is referenced again in line 244 right before Theorem 8). We will make sure the significance of Theorem 4 is well-highlighted.
>
> Q3. “The argument over how to test the dispersion assumption in the sequence of high-dimensional loss functions of graph parameters in practice is unclear.”
>
> A3. The main focus of the paper is to provide strong theoretical foundations for our approaches.
>
> It is true that at this point we don’t have a practical and provably correct procedure to test dispersion. However, we believe that this is not a major limitation of our work because of the following reasons:
>
> (1) The interesting aspect is that our algorithms do not need as input the dispersion parameters. The dispersion parameters just appear in our guarantees on the regret --  in particular, if the data is dispersed with good parameters, then our regret bound improves.
>
> (2) In our paper we also provide explicit useful sufficient conditions for dispersion to hold. We show that dispersion holds under natural smoothness assumptions about input data, for example distances between nodes have a smooth distribution (line 205-207) which could be easier to verify.
>
> (3) We would also like to point that in many other cases in learning theory and theory of computing, it is often hard to test data-dependent properties (e.g. Rademacher complexity for sample complexity or stability properties for computational efficiency of clustering procedures), nonetheless  there are intense efforts on designing algorithms that take advantage of such structure in data when it exists.
>
>
> Q4. “The dataset size, n, is maintained relatively modest (n=100) compared to the size of the benchmark datasets that were utilized, most likely due to the high time complexity of the optimization process for learning the graph parameters (MNIST, CIFAR, Omnigot).”
>
> A4. Note that the distribution (see Section 5) is considered over the problem instances (partially labeled graphs) as the 'data' and the 'hypothesis’ learned is a (semi-supervised) learning algorithm (instead of just learning a classifier over the label distribution for the graph nodes). The problem instances are related, but it does not imply or assume a common label distribution across instances. As a simple example, note that a large number of examples can help improve the accuracy for all parameters in the graph family, but learning which parameter performs best (on average across instances) is a distinct problem (may not depend on dataset size).
>
> Therefore our theoretical setting does not require large dataset sizes for validation. There are also some practical considerations for setting $n=100$ noted below.
>
>     -- Given a large instance/dataset we can always learn with small batches. Note that our choice of batch size $n$ is comparable to `minibatch` size that practitioners use in SGD (stochastic gradient descent) for deep learning which is typically even smaller than 100.
>
>     -- Also note that classic graph-based semi-supervised learning on a single instance is typically $O(n^3)$ or worse, our procedures have (e.g. Theorem 9) have a comparable time complexity. There is additional literature on scalable graph-based SSL, but it is significantly orthogonal to the contributions here.
>
> Q5. “The low generalization of the optimal parameter values determined via transductive training, as shown in the experimental part, might be explained by the limited sample size used in training.”
>
> A5. We disagree with the reviewer for several reasons:
>      -- Our online learning scenarios (Section 4), can model the challenging setting where  the data (whole problem instances we might want to solve) just comes online and the instances in the sequence are potentially  different.  The reviewer seems to miss the point about online learning that it is supposed to help when one is faced with unseen instances which can be different from ones seen before. “Low generalization” across instances can happen even for large instances because the problem instances are different, our claim is that our approaches obtain low regret (nearly match the performance of the parameter which performs best on average) even when this happens.
>
> A small sample size from the benchmark dataset therefore allows us to simulate this more challenging setting of “low generalization” across instances using a fixed large dataset. Additionally, practical motivations for choosing the instance size are noted below.
>
>      -- A useful analogy may be SGD using batches versus gradient descent over the full dataset (the former is better w.r.t. both efficiency and accuracy). Of course gradient-based optimization does not actually apply to our setting where the loss function is highly discontinuous.
>
>      -- As remarked above, using known techniques for graph-based SSL (even for a single fixed parameter) on full datasets can be computationally prohibitive. Determining the optimal parameter over a real-valued family with non-convex non-Lipschitz loss function is even harder.

---

### Official Review · Reviewer_HLTa · 2021-07-25

**Rating:** 8
**Confidence:** 3

**Summary:**

This work presents novel theoretical tools for data-driven algorithm design. In particular, the authors generalise a dispersion-based analysis from intervals to axis-aligned paths of arbitrary dimension.

Using these new results, the authors study graph-based semi-supervised learning from a data-driven perspective and achieve strong bounds in the online and PAC setting. Their goal is to perform well on a fixed distribution of semi-supervised problem instances (or minimising the regret in the online setting) by selecting the most appropriate graph from a parametrised graph family (in particular: threshold graphs and graphs with edge weights given by the polynomial or RBF kernel). Many additional results, such as first experiments and lower bounds, are discussed.

**Limitations And Societal Impact:**

The authors do not discuss any limitations. In my opinion, there are some topics, which the authors might address:
* Is it possible to get similar bounds for (one of the most commonly used similarity graph) $k$-nearest neighbour graphs ($k$-NNs)? Why do the authors have restricted their results to the other three types (threshold, polynomial, RBF)?
* Do the results still hold when using other classification algorithms, e.g., learning with local and global (Zhou, et al. 2004) or GCNs?
* This work mostly focuses on setting the edge-weights (or filtering edges) of the graph given feature-based distances. There might be scenarios where restricting the whole graph family (e.g., to trees) is more reasonable than just setting the weights.
* The double exponential dependence on $p$ in Theorem 3 seems to be problematic. Please, discuss why this is theoretically / asymptotically not an issue. Also mention this in the main text, instead of hiding the "constant" in the $O(\cdot)$. It seems to severely limit the practicability of these bounds.


The authors only briefly discuss societal impact. However, this is completely fine for such a conceptual and theoretical work

**Main Review:**

This is a very interesting and important work studying semi-supervised learning from a data-driven perspective. It opens up many interesting follow-up questions and makes one interested in reading more on that topic. It is an important contribution to the problem of learning / constructing the graph for semi-supervised learning.

Even though some parts (section 1, 2 and the first paragraph of section 4 with Theorem 5) are very well-written, the main technical sections of this work are quite dense and sometimes hard to follow. Notation and concepts are sometimes not well introduced and discussed. E.g., the $G(r)$ (and related) are not formally introduced.

Also, the proposed algorithms are only very briefly described. The notation and details are not really clear. Algorithm 2 is not referenced from the main text.

Some more related previous papers could be discussed in more depth, in particular previous theoretical results on graph construction such as
* Maier, Markus, Ulrike Von Luxburg, and Matthias Hein. "Influence of graph construction on graph-based clustering measures." NIPS. Vol. 1025. 2008.

and, for example, the discussions in:
* Liu, Wei, Junfeng He, and Shih-Fu Chang. "Large graph construction for scalable semi-supervised learning." ICML. 2010.
* Jebara, Tony, Jun Wang, and Shih-Fu Chang. "Graph construction and b-matching for semi-supervised learning." Proceedings of the 26th annual international conference on machine learning. 2009.

Some additional minor comments:

* line 41: $L+U$, why not use the common notation $L\cup U$.
* line 69: $G(\rho)$ where $\rho$ corresponds to a semi-supervised learning (SSL) algorithm. However later the authors rather use e.g., $G(r)$ or $G(\sigma)$ to denote the parameter to construct the graph and not the SSL algorithm itself. This is a little bit confusing. It would also help a lot to properly define $G(r)$, $G(\sigma)$, and $G(\tilde{\alpha})$,e.g., in Def. 1.
* line 103: Is it allowed to have the same examples in $L$ and $U$?
* In line 113, the authors call $d$ a similarity function (which would be large for similar examples), but seem to use it rather as a distance/metric (small for similar examples)
* line 116: Why not use the more commonly used term $\varepsilon$-neighbourhood graph ($\varepsilon$-NN) instead of "threshold graphs". The name threshold graphs is also used to describe a particular unweighted graph family.
* line 206: What embeddings do the authors mean here?
* The set $C$ used in algorithm 1 and 2 is not defined nor described. Algorithm 1: Two dots ".." in line 6. Missing "dp" in line 9.
* line 237: typo in "existin", -> "exist in"
* line 408: reference: missing "ö" in Schölkopf. Also: the year seems to be wrong, the book was published in 2006, and is the ISBN really required?
* a conclusion paragraph would be nice.

**Time Spent Reviewing:**

6

---

> ### Author Response · Authors · 2021-08-11
> **Response to Reviewer HLTa**
>
> Thank you for the review and the thoughtful feedback and suggestions! We are glad to hear that you find our work very interesting and important, and we agree that it opens up many interesting follow-up questions.
>
> We will address your suggestions and incorporate the related work in the final version of the paper. We provide concrete clarifications and answers to your questions below.
>
> 1. Notation:  We will clarify the notation as you suggest. Note that a general $G(\rho)$ is formally introduced in lines 107-108. We will include this as part of Definition 1 for better visibility as suggested.
>
> 2. About the choice of graph types: We consider threshold-based (e.g. $\epsilon$-neighborhood graphs, $G(r)$ in the paper's notation) and kernel-based (polynomial and RBF) families for studying data-driven optimization over the parameterized graphs. The $k$-NN family is a finite-parameter family, so online learning is possible by modeling using finite experts (e.g. “Online Learning and Online Convex Optimization”, Shalev-Shwartz 2011). This makes it significantly easier to give regret bounds compared to the continuous parameter families we consider, we are happy to add this remark to our paper (stated more technically at the following [link](https://drive.google.com/file/d/1xJ9WIHYPoWY1DZu3F6Q0Rx6dKPCxR-oZ/view)). In the final version we will add this additional popular family which can help increase the scope and appeal of our work. Thank you for the suggestion.
>
>     -- Other combinatorial constraints (e.g. restricting to trees) are not covered by our analysis but sound like interesting directions for future work, potentially extending impact beyond semi-supervised learning. We will add it as an interesting open direction.
>
> 3. Other classification algorithms: Some of our results (example Theorem 6 for threshold graphs $G(r)$) apply to any classification algorithm. For the vast majority of our results we can say the following.
>
>      -- Extension of our results to (Zhou, et al. 2004) is straightforward since the graph regularization constraint $H(f,W)$ is similar to the harmonic function objective and label propagation considered in the paper (there is an additional parameter $\alpha$ but the discontinuities are still located at zeros of polynomials, stated more technically at the following [link](https://drive.google.com/file/d/1xJ9WIHYPoWY1DZu3F6Q0Rx6dKPCxR-oZ/view)). The classification algorithm used in GCN is significantly different, and formal guarantees for this class of algorithms would likely need new insights. We will include discussion of these, and other related works mentioned by the reviewer, in the next version.
>
> 4. The 'double exponential' dependence on $p$ in the bounds used in the proof of Theorem 3 is actually not an issue as explained below. We will add the comment to the paper for clarity.
>
>      --  $p$ in our context of semi-supervised learning refers to number of distance metrics (e.g. in multimodal data) for the application so it is reasonable to think of it as a constant and discussing asymptotics in $p$ does not seem very relevant. We are happy to “unhide” this dependence.

---

> > ### Comment · Reviewer_HLTa · 2021-08-27
> > **Response to clarifications**
> >
> > Thanks for the clarifications and the comments on $k$-NN. I am now even more certain about the quality and importance of this work.

---

### Decision · Program_Chairs · 2021-09-27

**Decision:**

Accept (Oral)

**Comment:**

The reviewers generally consider this a strong theoretical submission that should be accepted despite a few shortcomings in the experiments. The author feedback clarified a few open questions and should be incorporated in the final version. The reviewers appreciated the solid theoretical work with impact to graph-based semi-supervised learning and beyond.